# Neural functional a posteriori error estimates

## Abstract

We propose a new loss function for supervised and physics-informed training of neural networks and operators that incorporates a posteriori error estimate. More specifically, during the training stage, the neural network learns additional physical fields that lead to rigorous error majorants after a computationally cheap post-processing stage. Theoretical results are based upon the theory of functional a posteriori error estimates, which allows for the systematic construction of such loss functions for a diverse class of practically relevant partial differential equations. From the numerical side, we demonstrate on a series of elliptic problems that for a variety of architectures and approaches (physics-informed neural networks, physics-informed neural operators, neural operators, and classical architectures in the regression and physics-informed settings), we can reach better or comparable accuracy and in addition to that cheaply recover high-quality upper bounds on the error after training.

## 1 Introduction

Recently it has been a surge of interest in cheap surrogates for partial-differential equation (PDEs) solvers in the machine learning (ML) community. This interest led to the construction of a plethora of novel efficient architectures including Fourier Neural Operator (FNO) Li et al. (2020), Deep Operator Network (DeelONet) Lu et al. (2019), wavelet- and transformer-based approaches Gupta et al. (2021), Hao et al. (2023).

Arguably, for low-dimensional case the most promising results appeared in the regression setting, when the neural network learns a mapping from the input PDEs data to the quantity of interest (e.g., the solution at a particular time or on the whole spacetime interval of interest), given observed data or results of the classical simulations. Often, it is possible to retain enough accuracy (the typical error is a few percent of the relative $L_2$ norm) and simultaneously reduce solution time by a few orders of magnitude Li et al. (2022), Brandstetter et al. (2022), Lam et al. (2022). In this scenario, one can hope to use neural network-based PDE solvers in compute-intensive applications such as weather forecasts, PDE-constraint optimization Biegler et al. (2003), or inverse problems Tarantola (2005).

Despite being promising, neural PDE solvers are unreliable. Even when the results on approximation capabilities are available Kovachki et al. (2021), Lanthaler et al. (2022), they do not guarantee that it is possible to reach good accuracy with standard training approaches (see Fokina & Oseledets (2023), Colbrook et al. (2022)). Since a priori error analysis seems futile, the natural alternative is a posteriori error analysis.[1] If reliable and computationally cheap a posteriori error analysis were possible, it would allow the safe application of neural PDE solvers.

With this note, we hope to draw the attention of the ML community to a well-developed powerful approach to a posteriori error analysis known as *functional a posteriori error analysis* Neittaanmäki & Repin (2004), Mali et al. (2013), Repin (2008), Muzalevsky & Repin (2021). The main strength

---

[1]We briefly recall the difference between a priori and a posteriori error analysis. The former one is an estimation that does not depend on the obtained solution, e.g., it can be a statement on the order of convergence: error $\leq Ch^p$, where $h$ is a grid spacing, $p \in \mathbb{N}$, and $C$ is an unspecified constant that depends on the problem data. The latter one is a concrete estimation that explicitly depends on the obtained approximate solution but not on the exact solution, i.e., error $\simeq f(\text{approximate solution, problem data})$.

of the technique is that it allows for error estimation regardless of the method used to construct approximation (contrast this with highly specialized approaches, e.g., FEM a posteriori analysis Ainsworth & Oden (1997)). This feature makes functional a posteriori analysis especially appealing for solvers based on neural networks that often act as black boxes.

More specifically, functional a posteriori error analysis supplies a technique to construct error majorant (upper bound on the error) that depends on approximate solution, PDE data, and additional fields that can be used to tighten the upper bound. The main properties of the functional approach are that: (i) the majorant remains upper bound for arbitrary approximate solution[2] regardless of the quality and the nature of the approximation, (ii) it provides a tight upper bound which is saturated for exact solution only. Using this powerful approach, we contribute the following:

1. In Section 2, following pioneering contribution Muzalevsky & Repin (2021), we define novel loss function for physics-informed training called Astral (neur**A**l a po**ST**erio**R**i function**A**l **L**oss).

2. In Section 3, we demonstrate that for elliptic equations training with Astral loss is much more robust than training with residual and variational losses (see Fig. 3). Besides that, Astral is equivariant to rescaling and allows for direct error control, which is impossible with other approaches.

3. As we explain in Section 2 Astral can also be used for parametric PDEs. We present two schemes to achieve that: the physics-informed (unsupervised) Fig. 1 and supervised Fig. 2. We test these schemes in Section 4 and find that in unsupervised setting Astral outperforms residual training used in PINO Li et al. (2021) by a significant margin (see Fig. 4).

## 2 FUNCTIONAL A POSTERIORI ERROR ESTIMATE AS A NOVEL LOSS FUNCTION

### 2.1 HIGH-LEVEL DESCRIPTION OF THE ERROR ESTIMATE

Consider PDE in the abstract form

$$\mathcal{A}\left[u, \mathcal{D}\right] = 0, \tag{1}$$

where $\mathcal{A}$ is a nonlinear operator containing partial derivatives of the solution $u$, $\mathcal{D}$ stands for supplementary data such as initial conditions, boundary conditions, and PDE parameters.

Typically, the exact solution to Eq. (1) is not available, so one obtains only approximate solution $\widetilde{u}$. Now, it is desirable to estimate the quality of $\widetilde{u}$, i.e., the distance between approximate and exact solution in some norm $\|\widetilde{u} - u_{\text{exact}}\|$. The estimate (i) should have definitive relation to the error norm (e.g., upper or lower bound), (ii) is computable only from PDE data and approximate solution $\widetilde{u}$, (iii) is cheaper to compute than the approximate solution $\widetilde{u}$ itself. Estimates following desiderata are known for specific discretizations, most notably for finite element methods Babuška & Rheinboldt (1978), but they are not applicable when the solution comes from neural networks. The only approach suitable for deep learning is discretization-agnostic functional a posteriori error analysis, which we now outline.

In general, functional a posteriori error bounds for a given PDE have a form

$$L[\widetilde{u}, \mathcal{D}, w_L] \leq \|\widetilde{u} - u_{\text{exact}}\| \leq U[\widetilde{u}, \mathcal{D}, w_U], \tag{2}$$

where $u$ is an approximate solution, $w_L$ and $w_U$ are arbitrary free functions (we call them *certificates* in the text below) from certain problem-dependent functional space, $\mathcal{D}$ is problem data (e.g., diffusion coefficient, viscosity, information on the geometry of the domain e.t.c.), $U$ and $L$, error majorant and minorant, are problem-dependent nonlinear functionals (see Section 2.3 for concrete example) Mali et al. (2013). Majorant and minorant has special properties:

1. They are defined in a continuous sense for arbitrary $\widetilde{u}$, $w_U$ from certain functional spaces, that is, they do not contain information on a particular solution method, grid quantities, convergence, or smoothness properties.

---

[2]The only requirement is that approximate solution should lay in the appropriate PDE-dependent functional space.

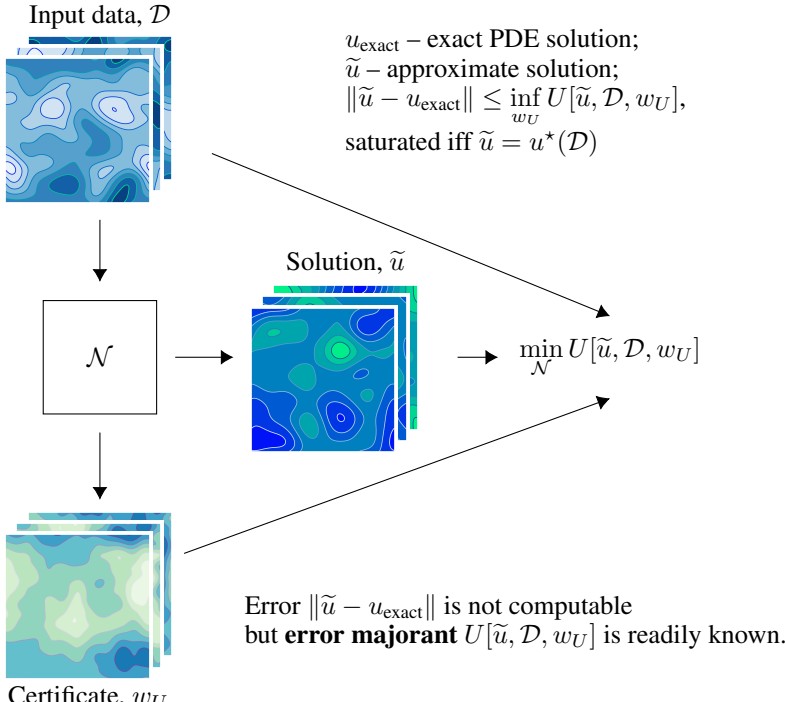

Figure 1: Unsupervised learning scheme with a posteriori functional error estimate and the explanation of basic properties of the upper bound. Neural network $\mathcal{N}$ takes input PDE data $\mathcal{D}$ and outputs approximate solution $\widetilde{u}$ and certificate $w_U$. Input to $\mathcal{N}$ and both outputs are plugged into **error majorant** $U[u, \mathcal{D}, w_U]$ which is used as an unsupervised loss.

2. The bounds are tight. That is, solving $\sup_{w_L, u} L[u, \mathcal{D}, w_L]$ or $\inf_{w_U, u} U[u, \mathcal{D}, w_U]$ one recovers exact solution $u_{\text{exact}}$.

3. They are explicitly computable based on $\widetilde{u}$ and the problem data $D$.

Typical application of Eq. (2) is to plug given approximate solution $\widetilde{u}$ and optimize for additional field $\sup_{w_L} L[\widetilde{u}, \mathcal{D}, w_L] \leq \|\widetilde{u} - u_{\text{exact}}\| \leq \inf_{w_U} U[\widetilde{u}, \mathcal{D}, w_U]$, to find error bounds as tight as possible. The optimization problems are potentially computationally demanding, so one can use deep learning to mitigate numerical costs.

The other possibility is to consider the upper bound in Eq. (2) as a novel loss function and optimize it for $u$ and $w_U$. Since the bounds are tight, $u$ converges to the exact solution, and since the majorant provides an upper bound, we have direct access to the upper bound on error during training.

We provide concrete examples of error bounds for elliptic equations and discuss how to construct them for other equations in Section 2.3, but before that we digress to explain in more details two approaches to combine deep learning with a posteriori functional error estimate outlined above.

## 2.2 FROM ERROR MAJORANT TO DEEP LEARNING

As we mentioned in the introduction, one of the most successful applications of deep learning to PDEs is the construction of cheap surrogates for classical solvers, i.e., the neural network produces an approximate solution given PDE data as input. To train such a network, one collects a dataset with pairs (PDE data, solution) using a classical solver and trains a neural network in the regression setting. One downside of this approach is the absence of guarantees. When neural networks provide solutions for unseen data, how can one measure the approximation error?

One possibility is to plug the solution $\widetilde{u}$ in discretization-agnostic upper bound $U[\widetilde{u}, \mathcal{D}, w_U]$ and optimize for $w_U$. This was done in Muzalevsky & Repin (2021) for solution obtained with physics-

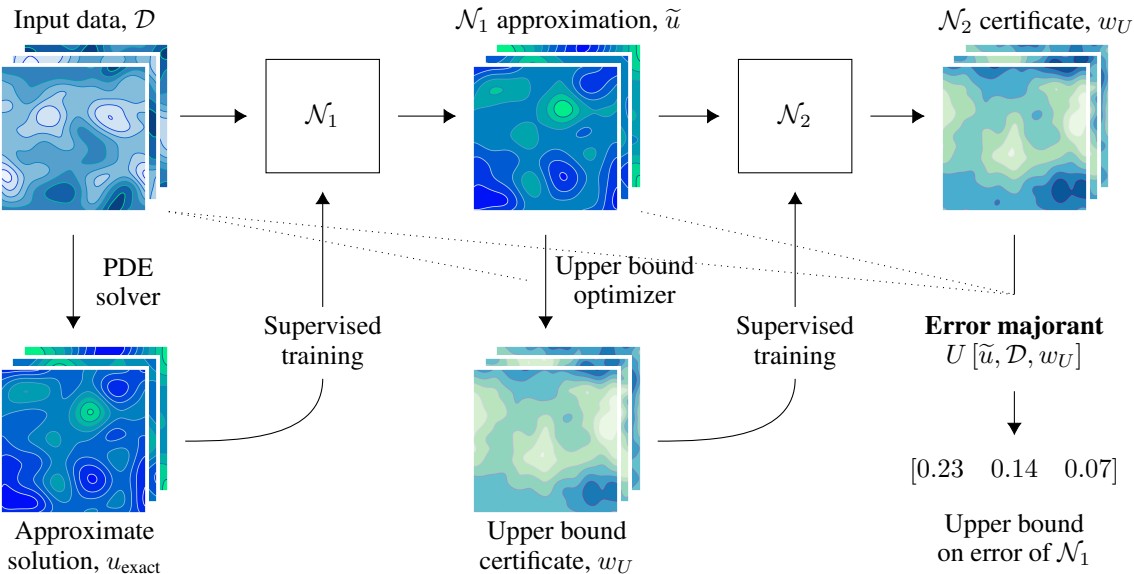

Figure 2: Supervised learning scheme with a posteriori functional error estimate. From left to right: (1) PDE is solved for selected input data; (2) pairs (input data, approximate solution) are used to train $\mathcal{N}_1$ with ordinary $L_2$ loss; (3) approximate solutions produced by $\mathcal{N}_1$ are plugged into upper bound and optimized certificates are found with direct optimization; (4) triples (input data, $\mathcal{N}_1$ approximation, certificate) are used to train $\mathcal{N}_2$ with ordinary $L_2$ loss; (5) certificates produced by $\mathcal{N}_2$, output of $\mathcal{N}_1$ and input data are plugged into upper bound to obtain bounds on error.

informed neural networks. However, the complexity of the optimization problem can be comparable with solving the original PDE, so this method does not look computationally appealing.

The natural alternative is find the solution $\widetilde{u}$ and the certificated $w_U$ in a single optimization run. For that we define novel loss function and the training strategy.

**Astral loss.** *For PDE with functional error majorant $U[\widetilde{u}, \mathcal{D}, w_U]$, and neural network $\mathcal{N}(\mathcal{D}, \theta) = (\widetilde{u}, w_U)$ with weights $\theta$ that predicts solution $\widetilde{u}$ and certificate $w_U$ optimize*

$$\min_{\theta} U[\widetilde{u}, \mathcal{D}, w_U] \ s.t. \ (\widetilde{u}, w_U) = \mathcal{N}(\mathcal{D}, \theta). \tag{3}$$

That is, training is done the same way as for classical physics-informed neural networks and neural operators, but with two differences. First, the loss function is the error majorant, not the residual. Second, the neural network has additional outputs since it needs to predict certificates $w_U$ along with the approximate solution $\widetilde{u}$. The concrete example of Astral loss is given in Section 2.3 for elliptic problem (see Eq. (6)), it is also explained there how to construct it for other PDEs.

Unfortunately, physics-informed approach are often computationally less efficient for low dimensional problems Grossmann et al. (2023), Karnakov et al. (2022). So one can resort to standard solvers and optimizers and simply replace the computationally demanding search for optimal $w_U$ with a cheap neural network surrogate. This may sound suspicious. One can argue that we will lose guarantees on the upper bound if we outsource optimization to possibly inaccurate neural networks. Fortunately, this is not the case since the upper bound holds for arbitrary $w_U$. When neural networks fail to predict sufficiently accurate $w_U$, the upper bound merely ceases to be tight but remains a genuine upper bound. These considerations lead to the approach summarized in Fig. 2. It consists of four stages:

1. Classical solver is used to produce dataset of pairs (PDE data, solution).

2. Neural network $\mathcal{N}_1$ is trained on the dataset to predict solutions from PDE data.

3. Predictions of neural network $\mathcal{N}_1$ are analyzed with the help of the upper bound, i.e., for each prediction certificate $w_U$ is obtained by direct optimization of the upper bound. From this data, a second dataset of triples (PDE data, solution, certificate $w_U$) is formed.

4. Second neural network $\mathcal{N}_2$ is trained to predict certificate $w_U$ from PDE data and approximate solution produced by $\mathcal{N}_1$.

At the inference stage, $\mathcal{N}_1$ predicts approximate solution $\widetilde{u}$ given problem data, $\mathcal{N}_2$ predicts certificate $w_U$ given approximate solution and problem data and the upper bound is used to compute upper bound on the error.

## 2.3 Concrete error bounds

To illustrate abstract approach described in Section 2.1 we consider elliptic equation in the domain $\Gamma \subseteq [0,1]^D$:

$$-\sum_{ij=1}^{D} \frac{\partial}{\partial x_i} \left( a_{ij}(x) \frac{\partial u}{\partial x_j} \right) + b^2(x)u(x) = f(x), \ u|_{\partial\Gamma} = 0, \ a_{ij}(x) \geq c > 0. \tag{4}$$

For later use we define several norms

$$\|v\|_2^2 = \int_\Gamma dx \, v^2, \ \|w\|_{a^{-1}}^2 = \int_\Gamma dx \left( \sum_{i,j} \left( a^{-1} \right)_{ij} w_i w_j \right), \ \|u\|^2 = \|u\|_a^2 + \|bu\|_2^2 . \tag{5}$$

Using the weak form, Cauchy-Schwarz, and Friedrichs inequalities, it is possible to show (see (Mali et al., 2013, Chapter 3) and also Appendix B) that for Eq. (4) the energy norm of the deviation of approximate solution $\widetilde{u}$ from the exact one $u$ is bounded from above:

$$\|\widetilde{u} - u_{\text{exact}}\|^2 \leq \mathcal{L}_{\text{Astral}}[\widetilde{u}, y, \mathcal{D}, \beta] = \int_\Gamma dx \frac{C^2(1+\beta)}{C^2 b(x)^2(1+\beta)+1} \mathcal{R}(\widetilde{u}, y)^2 + \frac{1+\beta}{\beta} \|a\nabla\widetilde{u} - y\|_{a^{-1}}^2 ,$$

$$\mathcal{R}(\widetilde{u}, y) = f(x) - b(x)^2\widetilde{u} + \sum_i \frac{\partial y_i(x)}{\partial x_i}, \ C = 1/\left( \inf_x \sqrt{\lambda_{\min}(a_{ij}(x))}\pi D \right),$$

$$\tag{6}$$

where $\beta$ is positive number, $y$ is a vector field. Note that upper bound is the Astral loss defined in Section 2.2 where $y$ and $\beta$ in Eq. (6) correspond to $w_U$ from the definition Eq. (3).

Eq. (6) possess properties described in Section 2. First, it is easy to see that the bound is computable from the problem data (for considered PDE Eq. (4) the data is $b(x)$, $a_{ij}(x)$, $f(x)$), approximate solution $\widetilde{u}(x)$ and supplementary parameters (vector field $y$ and $\beta$). Second, it is straightforward to check that the bound is tight, since it is saturated when $y(x) = \sum_{j=1}^{D} a_{ij} \frac{\partial}{\partial x_j} u_{\text{exact}}$ and $\beta \to \infty$. For the formal proofs we refer to (Mali et al., 2013, Chapter 3).

Our second example of functional a posteriori upper bound is given for initial-boundary value problem

$$\frac{\partial u(x,t)}{\partial t} - \frac{\partial^2 u(x,t)}{\partial x^2} + a\frac{\partial u(x,t)}{\partial x} = f(x,t), \ u(x,0) = \phi(x), \ u(x,t)|_{x\in\partial\Gamma} = 0, \ a = \text{const}, \tag{7}$$

where $\Gamma = [0,1] \times [0,T]$. As shown in Repin & Tomar (2010), the upper bound for this case reads

$$\|e\|_{\text{c.d.}} \equiv \int dxdt \left( \frac{\partial e}{\partial x} \right)^2 + \frac{1}{2} \int dx \ e^2|_{t=T} \leq \mathcal{L}_{\text{Astral}}[v, y, \mathcal{D}],$$

$$\mathcal{L}_{\text{Astral}}[v, y, \mathcal{D}] = \int dxdt \left( \left( y - \frac{\partial v}{\partial x} \right)^2 + \frac{1}{\pi} \left( f - \frac{\partial v}{\partial t} - a\frac{\partial v}{\partial x} + \frac{\partial y}{\partial x} \right)^2 \right), \tag{8}$$

where $e(x,t) = u(x,t) - v(x,t)$ is the error, $v(x,t)$ and $u(x,t)$ are approximate and exact solutions, $y(x,t)$ is a free field corresponding to $w_U$ from the definition Eq. (3).

We have seen that for elliptic and convection-diffusion equations, the bounds are available. It is also positive to derive similar bounds for other practically-relevant PDEs. In particular, bounds are known for Maxwell equations Repin (2007), reaction-diffusion problems Repin & Sauter (2006), elastoplasticity problems Repin & Valdman (2009), PDEs for the flow of viscose fluid Repin (2002), and non-linear elliptic problems Repin (1999).

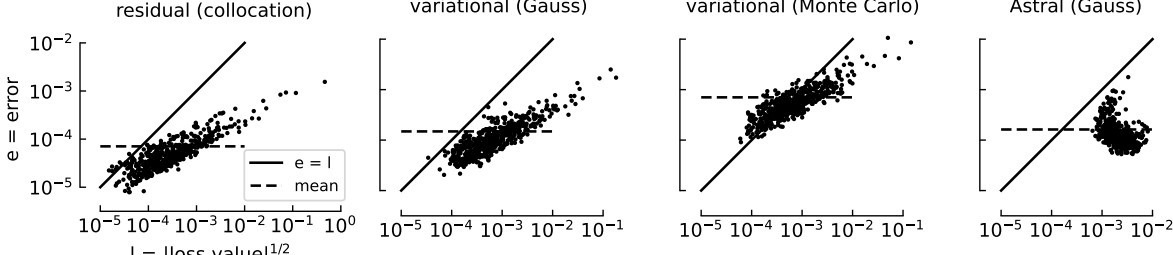

Figure 3: Comparison of PiNN solutions obtained with four different loss functions (from left to right): residual loss (Eq. (11), first equation) enforced on the set of points, variational loss (Eq. (11), second equation) computed with Gauss quadrature, variational loss computed with Monte Carlo, Astral loss Eq. (10) computed with Gauss quadrature. On each plot $x$-axis represents square root of the absolute value of the loss and $y$-axis represents the value of error in the energy norm $\||e\||$ (see Eq. (5)), both computed at the end of the optimization process.

Besides, there are a few general techniques to obtain functional error estimates. Most notable is the use of dual theory for variational problems with convex functionals Repin (2000a), Repin (2000b). More details on functional error estimates and systematic ways to derive them are available in monographs Repin (2008), Neittaanmäki & Repin (2004), Mali et al. (2013).

## 2.4 RELATION BETWEEN ENERGY AND $L_2$ NORMS

Since the $L_2$ norm is the most widely used one, it is instructive to establish how it is related to the energy norm used in the previous section. For example, for elliptic equations, we can find

$$\||e\||^2 = \int_\Gamma dx \sum_{ij} \frac{\partial e}{\partial x_i} \frac{\partial e}{\partial x_j} a_{ij} + \|be\|_2^2 \geq \lambda_{\min}^2 \|e\|_2^2 + \|be\|_2^2 \geq \lambda_{\min}^2 ||e||^2 + \inf_x b(x)^2 \|e\|_2^2, \quad (9)$$

where $\lambda_{\min}$ is a minimal eigenvalue of the elliptic problem $-\frac{\partial}{\partial x_i} a_{ij} \frac{\partial}{\partial x_j} u(x) = \lambda u(x)$ defined on the same domain and with the same boundary conditions as the original elliptic problem. From the expression above, we obtain the bound $\|e\|_2^2 \leq \frac{1}{\lambda_{\min}^2 + \inf_x (b(x))^2} \||e\||$. With the same reasoning, we can obtain a lower bound and find that $L_2$ and energy norms are equivalent (for suitably defined space of functions). The upper bound alone is sufficient to claim that whenever the error energy norm is sufficiently small, the $L_2$ norm of error is small too.

## 3 PiNN TRAINING WITH ASTRAL, RESIDUAL AND VARIATIONAL LOSSES

To practically evaluate Astral loss function, we compare it to other available losses for elliptic PDE. For simplicity we consider Eq. (4) with $a_{ij}(x) = a(x)\delta_{ij}$, $b(x) = 0$, $D = 2$, where $\delta_{ij}$ are elements of identity matrix. For this particular case, upper bound Eq. (6) simplifies (see Appendix C), and we

Table 1: Quality of the upper bound $\left(\sqrt{\mathcal{L}_{\text{Astral}}} - \||e\||\right)/\||e\||$ and comparison of error in two different norms for elliptic Eq. (4) in the $L-$shaped domain and convection-diffusion Eq. (7) equations.

| bound quality | elliptic, L-shaped domain 0.84 ± 0.60 | | convection-diffusion 0.63 ± 0.18 | |
|---|---|---|---|---|
| loss | relative error | energy norm | relative error | energy norm |
| Astral | $(7.3 \pm 9.8)\,10^{-3}$ | $(2.7 \pm 4.1)\,10^{-2}$ | $(8.3 \pm 8.9)\,10^{-3}$ | $(1.3 \pm 0.6)\,10^{-2}$ |
| residual | $(5.0 \pm 7.2)\,10^{-3}$ | $(1.4 \pm 1.2)\,10^{-2}$ | $(9.9 \pm 11)\,10^{-3}$ | $(7.8 \pm 2.4)\,10^{-3}$ |

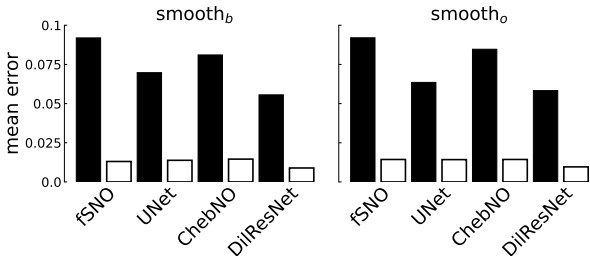

Figure 4: Comparison of errors for training with residual loss Eq. (11) (black) used in PINO Li et al. (2021) and Astral loss Eq. (10) (white) for elliptic equations with smooth coefficient. The results for Astral loss are better by large margin.

obtain a loss function

$$\mathcal{L}_{\text{Astral}}[u, y, \mathcal{D}, \beta] = (1 + \beta) \int dx \left( \frac{\left( f(x) + \sum_{i=1}^{2} \frac{\partial y_i}{\partial x_i} \right)^2}{4\pi^2 \inf_x a(x)} + \frac{\sum_{i=1}^{2} \left( a(x) \frac{\partial u}{\partial x_i} - y_i \right)^2}{\beta a(x)} \right). \quad (10)$$

For the elliptic equation, one can also apply standard residual and variational loss functions[3]

$$\mathcal{L}_{\text{res}} = \int dx \left( \sum_{i=1}^{2} \frac{\partial}{\partial x_i} \left( a \frac{\partial u}{\partial x_i} \right) + f \right)^2, \; \mathcal{L}_{\text{var}} = \int dx \left( \frac{1}{2} a(x) \sum_{i=1}^{2} \left( \frac{\partial u}{\partial x_i} \right)^2 - fu \right), \quad (11)$$

widely used in physics-informed neural networks Lagaris et al. (1998), Raissi et al. (2019), and Deep Ritz method E & Yu (2018).

To compare three loss functions, we produce a dataset for elliptic equations with known exact solutions and randomly generated smooth $a(x)$ with widely varying magnitudes (see Appendix D for details on dataset and training). On this dataset, we train physics-informed networks of similar size under the same optimization setting for three different losses above. To approximate integral, we use Monte Carlo (Kalos & Whitlock, 2009, Section 4) method or Gauss quadratures (Tyrtyshnikov, 1997, Lecture 16), and in case of residual loss, we used standard collocation formulation. In the network design and training, we closely follow the best practices for PiNN training Wang et al. (2023). A complete description of the training, network architectures, and dataset generation are available in Appendix J. To measure the quality of predicted solution we use error in the energy norm $\|\|e\|\|$ (see Eq. (5)).

In addition we perform two more experiments. Namely, training with residual and Astral loss Eq. (8) for convection-diffusion problem Eq. (7), as well as for the elliptic problem Eq. (4) in $L$-shaped domain, i.e., $\Gamma = [0, 1]^2 \backslash [0.5, 1]^2$. Details on these experiments are given in Appendix E and Appendix F respectively. The $L-$shaped domain is included as a classical benchmark used in literature on a posteriori error estimate.

### 3.1 DISCUSSION OF TRAINING RESULTS

Results for losses comparison appear in Fig. 3. We can make several important observations:

1. For all losses but $\mathcal{L}_{\text{Astral}}$, there are samples with errors larger and smaller than loss (there are samples from both sides on the line with slope equals one). This means only $\mathcal{L}_{\text{Astral}}$ provides the upper bound for the error.

2. Astral loss has a much smaller spread for the value of the loss and for the error, i.e., the optimization process with this loss is more robust. This can be explained by the fact, that Astral loss is equivariant to rescaling. Suppose we multiply elliptic equation (4) by $s$. It is easy to see that for Astral loss Eq. (10) one obtains overall scale $s^2$, i.e., the same scale

---

[3]For consistency, here we write residual loss in a continuous form, but in our experiments, we used the standard collocation approach.

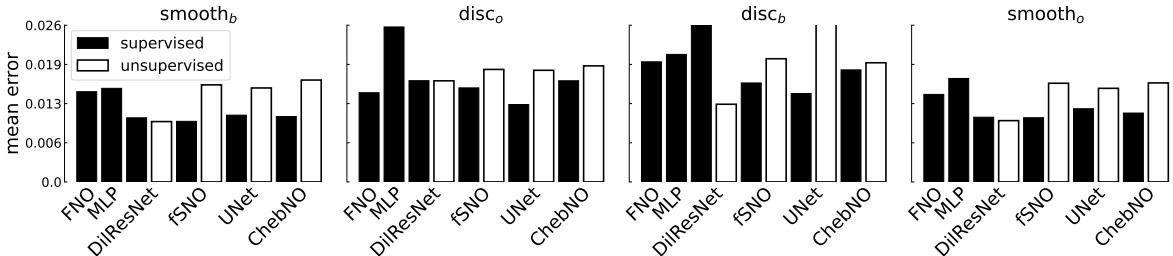

Figure 5: Comparison of mean errors in energy norm obtained with unsupervised (Fig. 1) and supervised (Fig. 2) training. Supervised training leads to slightly smaller error.

      as for squared error, simply by rescaling neural network output, i.e., $y_i \longrightarrow s y_i$. This is automatically done during training, so Astral loss converges more uniformly.

3. On average, we see that residual loss shows faster convergence. However, when the number of iterations is increased, we reach roughly the same errors with all losses.

The results for convection-diffusion equation and elliptic equation in the $L-$ shaped domain appear in Table 1. We can highlight the following:

1. Statistically, astral and residual losses typically provide roughly the same performance. On average, residual loss leads to slightly better relative error for elliptic problem and slightly worse for convection-diffusion equation.

2. For both equations Astral loss provide high-quality upper bound. This is especially interesting for $L-$shaped domain since solution has corner singularity.

## 4 APPLICATION TO PARAMETRIC PDES

In the case of parametric PDE, our experiments were guided by the following research questions: (i) What is the difference in accuracy between supervised (Fig. 2) and unsupervised (Fig. 1) settings, which one leads to tighter bounds on error? (ii) How sensitive is training to the parameters of PDE Eq. (4)? In particular, how well does the training work for discontinuous diffusion coefficients and zero source terms? (iii) In the unsupervised setting (Fig. 1), how does the upper bound loss compare with classical residual losses? (iv) How do the results depend on the choice of architecture?

To answer these questions, we generated four datasets with different source terms and diffusion coefficients. Namely, the equations with their short names are: $\text{smooth}_b - a(x)$ is smooth, $b(x)$ is present, (23); $\text{disc}_o - a(x)$ is discontinuous, $b(x)$ is zero, (24); $\text{disc}_b - a(x)$ is discontinuous, $b(x)$ is present, (25); $\text{smooth}_o - a(x)$ is smooth, $b(x)$ is zero, (26).

For these datasets we trained several architectures with different losses, including $L_2$ loss for the supervised training scheme (Fig. 2), upper bound loss Eq. (6) for $D = 2$ and the residual loss Eq. (11) for the unsupervised training scheme (Fig. 1). The architectures used are: (i) FNO – classical Fourier Neural Operator from Li et al. (2020); (ii) fSNO – Spectral Neural Operator on Gauss-Chebyshev grid. The construction mirrors FNO, but instead of FFT, a transformation based on Gauss quadratures is used Fanaskov & Oseledets (2022); (iii) ChebNO – Spectral Neural Operator on Chebyshev grid. Again, the construction is the same as for FNO, but DCT-I is used instead of FFT Fanaskov & Oseledets (2022); (iv) DilResNet – Dilated Residual Network from Yu et al. (2017), Stachenfeld et al. (2021); (v) UNet – classical computer vision architecture introduced in Ronneberger et al. (2015); (vi) MLP – vanilla feedforward neural network. So, we cover modern neural operators (FNO, fSNO, ChebNO) and classical computer vision architectures widely used for PDE modeling (DilResNet, UNet). MLP appears as a weak baseline.

To measure the quality of the predicted solution, we, again, use error in the energy norm $\|\|e\|\|$ (see Eq. (5)), and to measure the quality of predicted certificates $y$, $\beta$, we use the ratio of upper bound (the square root of the loss function Eq. (10)) to the error in the energy norm. The choice of metric for certificates is motivated by the fact that the only end of $y$ and $\beta$ is to provide an upper bound.

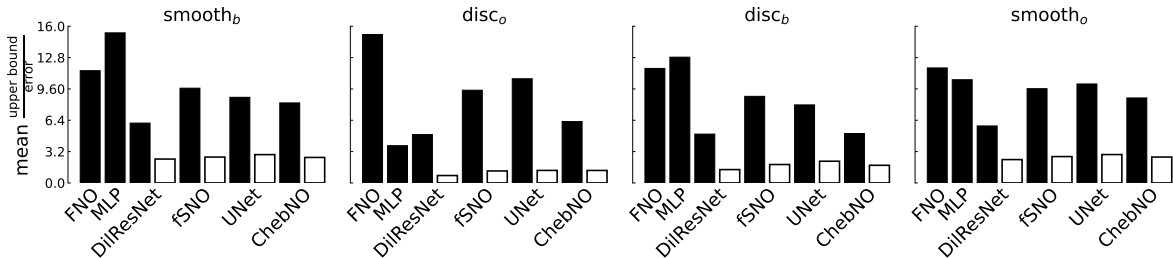

Figure 6: Comparison of relative upper bounds obtained with unsupervised (Fig. 1) and supervised (Fig. 2) training. Unsupervised training leads to much tighter error upper bound.

Also, only relative value is of interest since even when the error is large, neural network can provide a good upper bound, if it predicts the magnitude of this error accurately.

More details on experiments appear in Appendix H and Appendix I. More experimental results can be found in Appendix J

### 4.1 DISCUSSION OF TRAINING RESULTS

We can make the following important observations:

1. We can see from Fig. 4 that the Astral loss function produces much better error than the PINN loss Li et al. (2021) for all architectures considered and all equations with smooth coefficients.

2. Fig. 5 indicates that the supervised training scheme leads to slightly better errors for all architectures but DilResNet.

3. From Fig. 6, we can conclude that unsupervised training leads to a much tighter error bound than the supervised one. The upper bound obtained with Astral loss is almost the same as the one obtained with numerically expensive direct optimization of the upper bound. This shows one can produce high-quality upper bounds cheaply with a neural network.

4. We do not observe any sensitivity to the smoothness of coefficients of the elliptic equation, the presence of the source term, and the architecture used. In general, DilResNet performs better than other architectures, and neural operators seem to show second-best results, but the differences are not prominent.

## 5 CONCLUSION

Following Muzalevsky & Repin (2021), we outlined how a posteriori error estimate of functional type can be used to train physics-informed neural networks, and neural operators in supervised and unsupervised settings. As we argue, error majorants provide a novel systematic way to construct loss functions for physics-informed training. Our results indicate that for elliptic equations, the proposed loss leads to more stable training than classical residual and variational losses. Besides that, it allows for direct error control. For the parametric PDE, functional a posteriori error estimate allows for the first time to obtain practical error upper bound. That way, one can estimate the quality of the solution obtained by a black-box neural network. Predicting error majorants with neural networks looks especially appealing since one simultaneously retains guarantees (thanks to the superb properties of functional error estimate) and substantially decreases the computational load by removing the need to solve a complex optimization problem to obtain a tighter upper bound.

## 6 REPRODUCIBILITY

We use JAX Bradbury et al. (2018), Equinox Kidger & Garcia (2021) and Optax Babuschkin et al. (2020).

The anonymous version of GitHub repository with notebooks, architectures and training scripts is https://anonymous.4open.science/r/UQNO-23BC.

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

## A  RELATED RESEARCH

We can classify the related research articles into several categories.

The first one consists of generalization error analysis for physics-informed neural networks, e.g., De Ryck et al. (2022), Mishra & Molinaro (2022), De Ryck & Mishra (2022), Gonon et al. (2022), Jiao et al. (2021). In such contributions, authors show that it is possible to reach a given approximation accuracy with a neural network of a particular size. The bound on error involves unknown quantities, such as the norm of the exact solution. This line of work provides scaling arguments and establishes how expressive a particular architecture can be, but the results are unusable for actual error estimates.

The second line of work is on adaptive loss estimation for physics-informed neural networks. The examples include Wu et al. (2023), Zubov et al. (2021). In a standard physics-informed neural network, the set of collocation points is either randomly selected E & Yu (2018) or fixed from the start Raissi et al. (2019). Adaptive loss estimation aims to select points such that the residual is uniformly small. Typically, this tends to improve the accuracy and stability of training. The problem is it does not allow for error control since the relation between the residual and the error is not straightforward.

The next set of articles is related to the generalization bound for neural operators, e.g., Kovachki et al. (2021), Lanthaler et al. (2022). In analogy with physics-informed neural networks, the results usually established are bounds on network size sufficient to solve a given class of parametric PDEs. For the same reasons, with those results, it is not possible to estimate the actual error in a practical setting.

Besides that, there are several contributions directly related to a posteriori error estimation Guo & Haghighat (2022), Filici (2010), Hillebrecht & Unger (2022), Berrone et al. (2022), Minakowski & Richter (2023), Roth et al. (2022), Cai et al. (2020). In all these contributions, authors specialize in a particular classical error bound to deep learning PDE/ODE solvers. Namely, Filici (2010) adopts a well-known error estimation for ODEs, based on the construction of related problems with exactly known solution Zadunaisky (1976). Similarly, Hillebrecht & Unger (2022) uses well-known exponential bound on error that involves residual and Lipschitz constant Hairer et al. (1993) and applies a neural network to perform the residual interpolation. Similarly, contributions Guo & Haghighat (2022), Berrone et al. (2022) and Cai et al. (2020) are based on FEM posterior error estimates, and Minakowski & Richter (2023), Roth et al. (2022) are on dual weighted residual estimator Becker & Rannacher (2001).

## B  Derivation of Astral loss for elliptic equation

The discussion in this section follows (Mali et al., 2013, Chapter 3).

We consider the boundary value problem

$$-\operatorname{div} a\nabla u(x) + b(x)^2 u(x) = f(x), \quad x \subseteq \Gamma = [0,\,1]^D,$$
$$u\big|_{\partial\Gamma} = 0. \tag{12}$$

where $a$ is a symmetric matrix satisfying the condition $ax \cdot x \geq c|x|^2$, $\forall x \in \mathbb{R}^D$ and $b^2$ is a nonnegative function of $x$. The general solution $u(x)$ of (12) is defined by the integral identity

$$\int_\Gamma dx \Big[ a\nabla u(x) \cdot \nabla w(x) + b^2(x) u(x) w(x) \Big] = \int_\Gamma dx\, f(x) w(x), \tag{13}$$

that holds for every $w(x)$ from Sobolev space of square summable functions with square summable derivatives up to the order 1, $H^1([0,\,1]^D)$, with Dirichlet boundary conditions. Let $\widetilde{u}(x)$ approximate the exact solution $u(x)$ of the problem (12). By (13) we deduce the relation

$$\int_\Gamma dx \Big[ a\nabla(u - \widetilde{u}) \cdot \nabla w + b^2(u - \widetilde{u})w \Big] = \int_\Gamma dx \Big[ fw - b^2\widetilde{u}w - \nabla\widetilde{u} \cdot \nabla w \Big].$$

Since $w$ vanishes at the boundary:

$$\int_\Gamma dx (\nabla \cdot (yw)) = yw\big|_{\partial\Gamma} = 0,$$

where $y(x) \in \mathbb{R}^D$ is arbitrary vector. We can rewrite previous relation as follows:

$$\int_\Gamma dx \Big[ a\nabla(u - \widetilde{u}) \cdot \nabla w + b^2(u - \widetilde{u})w \Big] = \int_\Gamma dx \Big[ \mathcal{R}(\widetilde{u},\,y)w + (y - a\nabla\widetilde{u}) \cdot \nabla w \Big], \tag{14}$$

where $\mathcal{R}(\widetilde{u}, y) = f - b^2\widetilde{u} + \operatorname{div} y$. Let us represent the first integral on the right-hand side of (14) as follows

$$\int_\Gamma dx\, \mathcal{R}(\widetilde{u},\,y)w = \int_\Gamma dx\, \alpha\mathcal{R}(\widetilde{u},\,y)w + \int_\Gamma dx\,(1 - \alpha)\mathcal{R}(\widetilde{u},\,y)w,$$

where $\alpha \in L^\infty_{[0,\,1]}(\Gamma) = \big\{ \alpha \in L^\infty(\Gamma) \,\big|\, 0 \leq \alpha(x) \leq 1 \big\}$ is a weight function. It is easy to see that

$$\int_\Gamma dx (y - a\nabla\widetilde{u}) \cdot \nabla w \leq \big\| y - a\nabla\widetilde{u} \big\|_{a^{-1}} \|\nabla w\|_a,$$

$$\int_\Gamma dx\, \mathcal{R}(\widetilde{u},\,y)w \leq C \big\| \mathcal{R}(\widetilde{u},\,y) \big\|_2 \|\nabla w\|_a,$$

where $C$ is a constant in the inequality

$$\|w\|_2 \le C\|\nabla w\|_a, \quad \forall w \in H^1\big([0,\,1]^D\big).$$

Then, we have

$$\left|\int_\Gamma dx\, \mathcal{R}(\widetilde{u},\,y)w\right| \le \left\|\frac{\alpha}{b}\mathcal{R}(\widetilde{u},\,y)\right\|_2 \|bw\|_2 + C\left\|(1-\alpha)\mathcal{R}(\widetilde{u},\,y)\right\|_2 \|\nabla w\|_a,$$

$$\|w\|_a^2 = \int_\Gamma dx\, aw\cdot w, \quad \|w\|_{a^{-1}}^2 = \int_\Gamma dx\, a^{-1}w\cdot w. \tag{15}$$

By setting $w = u - \widetilde{u}$ we arrive at the estimate

$$\||\widetilde{u} - u\||^2 \le \left(C\left\|(1-\alpha)\mathcal{R}(\widetilde{u},\,y)\right\|_2 + \left\|a\nabla\widetilde{u} - y\right\|_{a^{-1}}\right)^2 + \left\|\frac{\alpha}{b}\mathcal{R}(\widetilde{u},\,y)\right\|_2^2.$$

In the sake of simplicity, we use the algebraic Young's inequality. For $a,\ b \in \mathbb{R}$ and for any $\beta$ positive number:

$$2ab \le \beta a^2 + \frac{1}{\beta}\, b^2.$$

We find that

$$\big|a + b\big|^2 \le \big(1 + \beta\big)\big|a\big|^2 + \frac{1+\beta}{\beta}\big|b\big|^2.$$

Thus, we have

$$\||\widetilde{u} - u\||^2 \le (1+\beta)C^2\left\|(1-\alpha)\mathcal{R}(\widetilde{u},\,y)\right\|_2^2 + \frac{1+\beta}{\beta}\left\|a\nabla\widetilde{u} - y\right\|_{a^{-1}} + \left\|\frac{\alpha}{b}\mathcal{R}(\widetilde{u},\,y)\right\|_2^2, \tag{16}$$

where $\beta$ is an arbitrary positive number.

Minimization of the right-hand side of (16) with respect to $\alpha$ is reduced to the following auxiliary variational problem: find $\widehat{\alpha} \in L^\infty_{[0,\,1]}(\Gamma)$ such that

$$\Upsilon\big(\widehat{\alpha}\big) = \inf_{\alpha \in L_{[0,\,1]}\infty(\Gamma)} \int_\Gamma dx\, \Big(\alpha^2 P(x) + \big(1 - \alpha\big)^2 Q(x)\Big),$$

and $P$ and $Q$ are nonnegative integrable functions, which do not vanish simultaneously. It is easy to find that for almost all $x$

$$\widehat{\alpha}(x) = \frac{Q}{P+Q} \in [0,\,1], \quad \Upsilon\big(\widehat{\alpha}\big) = \frac{PQ}{P+Q}.$$

In our case, $P = b^{-2}\mathcal{R}\big(\widetilde{u},\,y\big)$ and $Q = C^2\big(1 + \beta\big)\mathcal{R}\big(\widetilde{u},\,y\big)$.

Therefore, we obtain

$$\||\widetilde{u} - u\||^2 \le \int_\Gamma dx \frac{C^2(1+\beta)}{C^2 b(x)^2(1+\beta)+1}\mathcal{R}(\widetilde{u},y)^2 + \frac{1+\beta}{\beta}\left\|a\nabla\widetilde{u} - y\right\|_{a^{-1}}^2 = \mathcal{L}_{\text{Astral}}[\widetilde{u}, y, \mathcal{D}, \beta],$$

$$\mathcal{R}(\widetilde{u}, y) = f(x) - b(x)^2\widetilde{u} + \text{div}\, y, \quad C = 1\Big/\left(\inf_x \sqrt{\lambda_{\min}(a)}\pi D\right). \tag{17}$$

## C  SIMPLIFIED ASTRAL LOSS FOR SCALAR DIFFUSION COEFFICIENT

In this section, we illustrate how to derive Eq. (10) from Eq. (6).

First, observe that $b = 0$ so the residual weight simplifies to $C^2(1 + \beta)$ and the residual itself becomes $f(x) - \sum_{i=1}^2 \frac{\partial y_i}{\partial x_i}$ for $D = 2$ problem considered. Next, the second term with flux condition simplifies because $a_{ij}(x) = \delta_{ij}a(x)$ where $\delta_{ij}$ are matrix elements of the identity matrix. More specifically, the inverse matrix $a^{-1}$ is simply a division by $a(x)$, so the term becomes $\frac{1}{a(x)}\sum_{i=1}^2\left(a(x)\frac{\partial\widetilde{u}(x)}{\partial x_i} - y_i\right)^2$. Finally, the constant $C$ simplifies to $1\Big/\left(2\pi\inf_x\sqrt{a(x)}\right)$, since for diagonal matrix $a_{ij}(x)$ the minimal eigenvalue is $a(x)$.

With all that we obtain a simplified version of Astral loss Eq. (10).

## D    EXPERIMENT 1: COMPARISON OF LOSSES FOR PHYSICS-INFORMED NEURAL NETWORKS

In all experiments, we use feedforward physics-informed neural networks with 3 layers, 50 features in each layer, and GELU activation functions. Following suggestions in Wang et al. (2023), we transform input coordinates to obtain 50 Fourier features. Each network was optimized for 50000 epoch with Lion optimizer Chen et al. (2023). Starting learning rate is $10^{-4}$ with exponential decay $\times 0.5$ each 10000 epoch. The output of the neural network was multiplied by $\sin(\pi x)\sin(\pi y)$ to enforce boundary conditions. When error majorant is used as loss functions, we predict $y_1$, $y_2$, and $u$ with three separate neural networks.

To generate random elliptic equation we draw random function from $f \sim \mathcal{N}\left(0, (I-\Delta)^{-2}\right)$ after that this function is shifter and rescaled such that $\min_x f(x) = 1, \max_x f(x) = 6$. After that we multiply this function on $1/s$ where $s$ is drawn from exponential distribution with mean 100. This way we obtain positive functions of widely varying scale that we use as diffusion coefficient $a(x)$. For all elliptic equations we fix exact solution to be $u(x) = x_1(1-x_1)x_2(1-x_2)$ and find $f(x)$ from known $a(x)$ and $u(x)$.

## E    EXPERIMENT 2: CONVECTION-DIFFUSION PROBLEM

The architecture and training scheme are the same as in Appendix D, but we train for 10000 epoch and multiply learning rate by 0.5 each 3000 epoch.

To test PiNN networks we sample exact solutions of convection-diffusion equation Eq. (7) given by

$$u(x,t) = \text{Re}\left(\sum_{k=0}^{N} c_k e^{-(2\pi k)^2 t - 2\pi k a i t + 2\pi k x i}\right)\sin(\pi x) = \widetilde{u}(x,t)\sin(\pi x). \tag{18}$$

It is easy to see that

$$\frac{\partial \widetilde{u}(x,t)}{\partial t} - \frac{\partial^2 \widetilde{u}(x,t)}{\partial x^2} + a\frac{\partial \widetilde{u}(x,t)}{\partial x} = 0, \tag{19}$$

and that $u(x,t)$ fulfills boundary conditions. If we choose

$$f(x,t) = \widetilde{u}(x,t)\left(\pi^2\sin(\pi x) + a\pi\cos(\pi x)\right) - 2\pi\frac{\partial \widetilde{u}(x,t)}{\partial x}\cos(\pi x), \tag{20}$$

than $u(x,t)$ solves original equation Eq. (7).

So, to generate exact solutions we used Eq. (18) with $c_k$ sampled from $(x+iy)/(1+(\pi k/5)^2)^2$ where $x, y \sim \mathcal{N}(0,1)$ and $a$ sampled from $10^{-2}\mathcal{N}(0,1)$, $N = 150$. After that we compute source term using Eq. (20).

## F    EXPERIMENT 3: ELLIPTIC EQUATION, L-SHAPED DOMAIN

The architecture and training scheme are the same as in Appendix E.

For this particular geometry exact solutions are available only for specific boundary conditions, so we use finite difference discretization to obtain solution on the fine grid and use this solution as a ground truth.

Parameters of the problem are sampled as follows

$$a(x) = (10g_1(x))^2 + 1, \ f(x) = 10g_2(x), \ b(x) = 0, \ g_1(x), g_2(x) \sim \mathcal{N}\left(0, \left(I - 10^{-1}\Delta\right)^{-2}\right), \tag{21}$$

where Laplace operator has periodic boundary conditions.

To enforce Dirichlet boundary conditions in all experiments we used mean value coordinates Floater (2003) as explained in Sukumar & Srivastava (2022).

## G   ELLIPTIC EQUATIONS USED FOR EXPERIMENTS WITH PARAMETRIC PDES

All elliptic PDEs considered are based on general form, described in (4).

First, we define random trigonometric polynomials

$$\mathcal{P}(N_1, N_2, \alpha) = \left\{ f(x) = \mathcal{R}\left( \sum_{m=0}^{N_1} \sum_{n=0}^{N_2} \frac{c_{mn} \exp\left(2\pi i(mx_1 + nx_2)\right)}{(1 + m + n)^\alpha} \right) : \mathcal{R}(c), \mathcal{I}(c) \simeq \mathcal{N}(0, I) \right\}.$$
(22)

For the first equation, we use Cholesky factorization to define matrix $a$ and random trigonometric polynomials for $b$ and $f$:

$$a(x) = \begin{pmatrix} \alpha(x) & 0 \\ \gamma(x) & \beta(x) \end{pmatrix} \begin{pmatrix} \alpha(x) & \gamma(x) \\ 0 & \beta(x) \end{pmatrix},$$

$$\alpha(x), \beta(x) \simeq 0.1\mathcal{P}(5, 5, 2) + 1;\ \gamma(x), b(x), f(x) \simeq \mathcal{P}(5, 5, 2).$$
(23)

The next equation has a discontinuous scalar diffusion coefficient:

$$a(x) = \alpha(x)I,\ \alpha(x) = \begin{cases} 10, & p_1(x) \geq 0; \\ 1, & p_1(x) < 0, \end{cases} \quad b(x) = 0,\ f(x) = 1,\ p_1(x) \simeq \mathcal{P}(5, 5, 2).$$
(24)

The analogous equation is dubbed "Darcy flow" in Li et al. (2020).

Third equation is similar to (24) but with more diverse $b$ and $f$:

$$a(x) = \alpha(x)I,\ \alpha(x) = \begin{cases} 10, & p_1(x) \geq 0; \\ 1, & p_1(x) < 0, \end{cases} \quad b(x),\ f(x),\ p_1(x) \simeq \mathcal{P}(5, 5, 2).$$
(25)

Last equation is similar to (23) but with $b = 0$:

$$a(x) = \begin{pmatrix} \alpha(x) & 0 \\ \gamma(x) & \beta(x) \end{pmatrix} \begin{pmatrix} \alpha(x) & \gamma(x) \\ 0 & \beta(x) \end{pmatrix},$$

$$\alpha(x), \beta(x) \simeq 0.1\mathcal{P}(5, 5, 2) + 1;\ \gamma(x), f(x) \simeq \mathcal{P}(5, 5, 2);\ b(x) = 0.$$
(26)

## H   EXPERIMENT 4: SUPERVISED TRAINING WITH BUILD-IN ERROR ESTIMATE

We performed 6 training runs with $N_{\text{train}} = 200, 400, 600, 800, 1000, 1200$ for all architectures listed in Section 4 and all equations. The number of grid points is $2^5 + 1$ along each dimension. To estimate the exact solution, we solve the same problem with a higher resolution grid, using $2^7 + 1$ points. The averaged results are reported in Tables 10, 11, 12, 14.

In addition, for the first equation (23), we gather data on the same 6 training runs with varying resolution $2^J + 1$, $J = 5, 6, 7$. For this experiment, averaged results appear in Table 13.

In all optimization runs, we train for 1000 epochs with Adam optimizer having the learning rate $10^{-3}$ multiplied by 0.5 each 200 epochs (exponential decay) and weight decay $10^{-2}$.

Details on architectures used are as follows.

1. Construction of FNO closely follows the one given in Li et al. (2020). We use 24 featrures in processor, 4 layers and the number of modes is $\lceil N_{\text{spatial}}/4 \rceil$.

2. Construction of fSNO is similar to the one of FNO (i.e., encoder-processor-decoder and integral kernel in place of the linear layer), but following Fanaskov & Oseledets (2022) we replace Fourier basis with orthogonal polynomials in the integral kernel. For this particular architecture, the construction of the integral kernel is as follows. We use the Gauss-Chebyshev grid, and compute projection on the basis of polynomials using Gauss quadratures Golub & Welsch (1969). After the projection on the space of polynomials, we apply three convolutions with kernel size 3. Finally, to return to the physical space, we compute sum $\sum_n c_n p_n(x)$ on the Gauss-Chebyshev grid, where $c_n$ are coefficients obtained after convolutions. In this case, the number of features in the processor is 34, the number of layers is 4 and the number of modes is $\lceil N_{\text{spatial}}/4 \rceil$

3. ChebNO is a spectral neural operator Fanaskov & Oseledets (2022) defined on Chebyshev grid. Construction of integral kernel is similar to fSNO but DCT is used to find projection on the polynomial space in place of Gauss quadratures. In the current architecture, the processor has 32 features, the number of layers is 4, and 16 modes are used for all grids.

4. DilResNet that we use closely follows architecture described in Stachenfeld et al. (2021). Namely, we use a processor with 24 features that has 4 layers. Each layer consists of convolutions with strides $[1, 2, 4, 8, 4, 2, 1]$, kernel size 3, and skip connection.

5. For UNet Ronneberger et al. (2015) we start with 10 features and double the number of features with each downsampling that decreases the number of grid points by the factor of 2 in each dimension. On each grid we use 2 convolutions (kernel size 3) and max pooling, transposed convolution are used for upsampling, and 3 convolutions (kernel size 3) appears on each grid after upsampling. In total, we have 4 grids.

6. MLP that we use consists of linear layers that process each dimension (including the feature dimension) separately. That way, the linear operator is defined by three matrices in $D = 2$. MLP uses 64 in the processor and has 4 layers.

For all networks, we use ReLU nonlinearity. Typical number of parameters for each network is given in the table below.

| | FNO | fSNO | ChebNO | DilResNet | UNet | MLP |
|---|---|---|---|---|---|---|
| # parameters | $668 \times 10^3$ | $130 \times 10^3$ | $115 \times 10^3$ | $147 \times 10^3$ | $248 \times 10^3$ | $24 \times 10^3$ |

## I  EXPERIMENT 5: UNSUPERVISED TRAINING WITH BUILD-IN ERROR ESTIMATE

The unsupervised case is addressed by analyzing the $D = 2$ of (4) and optimizing the upper bound (6) to produce an approximate solution $\widetilde{u}$ and an upper bound certificate $y$. The loss that was used to train the neural network $\mathcal{N}$ is:

$$\mathcal{L}[\widetilde{u}, y, \mathcal{D}, \beta, \lambda] = \sqrt{\mathcal{L}_{\text{Astral}}[\widetilde{u}, y, \mathcal{D}, \beta]} + \lambda\sqrt{\widetilde{u}_{\partial\Gamma}^2}, \qquad (27)$$

$$\mathcal{L}[\widetilde{u}, y, \mathcal{D}, \beta, \lambda] \to \min_{\widetilde{u},\, y},$$

where $\widetilde{u},\, y$ are the output of the neural network, $a, b, f, C$ are defined from problem data $\mathcal{D}$ for each equation, and $\lambda$ and $\beta$ are hyperparameters of the loss function (27). Including the part of the loss with boundaries is necessary to compensate for the lack of information about the exact solution on $x \in \partial\Gamma$.

In this paper, a comparison was made with the state-of-the-art unsupervised model proposed by Li et al. (2021). The PINO loss is defined as

$$\mathcal{L}_{\text{pino}}[\widetilde{u}, u_{\text{exact}}, \mathcal{D}, \alpha, \gamma] = \mathcal{L}_{\text{data}}[\widetilde{u}, u_{\text{exact}}] + \alpha\mathcal{L}_{\text{residual}}[\widetilde{u}, \mathcal{D}] + \gamma\sqrt{\widetilde{u}_{\partial\Gamma}^2}, \qquad (28)$$

where

$$\mathcal{L}_{\text{residual}}[\widetilde{u}, \mathcal{D}] = \sqrt{\int_\Gamma dx \left(\sum_{i=1}^2 \frac{\partial}{\partial x_i}\left(a(x)\frac{\partial\widetilde{u}}{\partial x_i}\right) + f(x) - b(x)\widetilde{u}\right)^2},$$

$$\mathcal{L}_{\text{data}}[\widetilde{u}, u_{\text{exact}}] = \sqrt{\int_\Gamma dx \left(\widetilde{u} - u_{\text{exact}}\right)^2}.$$

The PINO loss consists of the physics loss in the interior and the data loss on the boundary condition, with hyperparameters $\alpha, \gamma > 0$. It is important to note that the proposed loss function (27) does not contain an exact solution $u_{\text{exact}}$ obtained using traditional solvers, unlike the PINO loss function (28).

We trained for 500 epochs with Adam optimizer that has the learning rate $2 \cdot 10^{-3}$ multiplied by 0.5 for each 50 epoch and weight decay $10^{-2}$. Hyperparameters $\lambda$ and $\beta$ were chosen to be 1.

The experiments are carried out according to the scheme in Fig. 1. We performed 9 training runs with $N_{\text{train}} = 200, 400, 600, 800, 1000, 1200, 1400, 1600, 1800$ for all architectures and all equations. The number of points in the grid is $2^5 + 1$ along each dimension. To estimate the exact solution, we solve the same problem with a higher resolution grid, using $2^7 + 1$ points. The results using our loss function (27) are presented in Tables 2,6,7,8,9. The results for PINO loss proposed in Li et al. (2021) are presented in Table 3.

The architectures used for unsupervised learning are identical to those used for supervised learning. For all networks, we use ReLU nonlinearity.

In Tables 15, 16, 17, 18, one can find the results for train runs with different $\lambda$ for all architectures and datasets for 2D equations. To check dependencies from $\lambda$, we used $N_{\text{train}} = 1200$ for training, and $N_{\text{test}} = 800$ for testing.

## J  ADDITIONAL DATA FOR EXPERIMENTS 2 AND 3

Table 2: Results for $N_{\text{train}} = 1800$, $N_{\text{test}} = 200$ for 2D equations using loss (27).

| equation | fSNO | | ChebNO | | DilResNet | | UNet | |
|---|---|---|---|---|---|---|---|---|
| | $E_{\text{train}}$ | $E_{\text{test}}$ | $E_{\text{train}}$ | $E_{\text{test}}$ | $E_{\text{train}}$ | $E_{\text{test}}$ | $E_{\text{train}}$ | $E_{\text{test}}$ |
| (23) | 0.012 | 0.013 | 0.013 | 0.015 | 0.008 | 0.009 | 0.012 | 0.014 |
| (24) | 0.019 | 0.018 | 0.019 | 0.018 | 0.017 | 0.016 | 0.025 | 0.024 |
| (25) | 0.014 | 0.017 | 0.015 | 0.018 | 0.011 | 0.012 | 0.070 | 0.078 |
| (26) | 0.013 | 0.014 | 0.013 | 0.014 | 0.009 | 0.010 | 0.013 | 0.014 |

Table 3: Results for $N_{\text{train}} = 1800$, $N_{\text{test}} = 200$ for 2D equations using PINO loss (28).

| equation | fSNO | | ChebNO | | DilResNet | | UNet | |
|---|---|---|---|---|---|---|---|---|
| | $E_{\text{train}}$ | $E_{\text{test}}$ | $E_{\text{train}}$ | $E_{\text{test}}$ | $E_{\text{train}}$ | $E_{\text{test}}$ | $E_{\text{train}}$ | $E_{\text{test}}$ |
| (23) | 0.091 | 0.092 | 0.081 | 0.081 | 0.055 | 0.056 | 0.065 | 0.070 |
| (26) | 0.091 | 0.092 | 0.082 | 0.085 | 0.058 | 0.058 | 0.061 | 0.064 |

Table 4: Results for distinct $N_{\text{train}}$ averaged with respect to equations for 2D using loss (27), $N_{\text{test}} = 200$.

| $N_{\text{train}}$ | fSNO | | ChebNO | | DilResNet | | UNet | |
|---|---|---|---|---|---|---|---|---|
| | $E_{\text{train}}$ | $E_{\text{train}}^{\text{up}}$ | $E_{\text{train}}$ | $E_{\text{train}}^{\text{up}}$ | $E_{\text{train}}$ | $E_{\text{train}}^{\text{up}}$ | $E_{\text{train}}$ | $E_{\text{train}}^{\text{up}}$ |
| 200 | 0.037 | 0.062 | 0.039 | 0.067 | 0.025 | 0.049 | 0.049 | 0.080 |
| 400 | 0.024 | 0.044 | 0.027 | 0.047 | 0.017 | 0.030 | 0.031 | 0.059 |
| 600 | 0.021 | 0.038 | 0.022 | 0.040 | 0.014 | 0.024 | 0.045 | 0.072 |
| 800 | 0.018 | 0.032 | 0.019 | 0.035 | 0.013 | 0.020 | 0.031 | 0.056 |
| 1000 | 0.017 | 0.029 | 0.018 | 0.032 | 0.013 | 0.018 | 0.030 | 0.054 |
| 1200 | 0.016 | 0.028 | 0.017 | 0.029 | 0.012 | 0.015 | 0.029 | 0.052 |
| 1400 | 0.015 | 0.026 | 0.016 | 0.028 | 0.012 | 0.014 | 0.028 | 0.051 |
| 1600 | 0.015 | 0.025 | 0.015 | 0.027 | 0.012 | 0.013 | 0.029 | 0.052 |
| 1800 | 0.014 | 0.024 | 0.015 | 0.026 | 0.011 | 0.012 | 0.030 | 0.053 |

Table 5: Results for distinct $N_{\text{train}}$ averaged with respect to equations for 2D using loss (27), $N_{\text{test}} = 200$.

| $N_{\text{train}}$ | fSNO | | ChebNO | | DilResNet | | UNet | |
|---|---|---|---|---|---|---|---|---|
| | $E_{\text{test}}$ | $E_{\text{test}}^{\text{up}}$ | $E_{\text{test}}$ | $E_{\text{test}}^{\text{up}}$ | $E_{\text{test}}$ | $E_{\text{test}}^{\text{up}}$ | $E_{\text{test}}$ | $E_{\text{test}}^{\text{up}}$ |
| 200 | 0.041 | 0.078 | 0.042 | 0.081 | 0.029 | 0.063 | 0.055 | 0.115 |
| 400 | 0.027 | 0.057 | 0.029 | 0.059 | 0.019 | 0.039 | 0.038 | 0.080 |
| 600 | 0.023 | 0.048 | 0.024 | 0.050 | 0.015 | 0.030 | 0.048 | 0.085 |
| 800 | 0.020 | 0.042 | 0.021 | 0.044 | 0.014 | 0.025 | 0.034 | 0.068 |
| 1000 | 0.018 | 0.038 | 0.020 | 0.040 | 0.013 | 0.022 | 0.033 | 0.066 |
| 1200 | 0.018 | 0.036 | 0.018 | 0.036 | 0.012 | 0.019 | 0.032 | 0.063 |
| 1400 | 0.017 | 0.033 | 0.018 | 0.034 | 0.012 | 0.017 | 0.031 | 0.059 |
| 1600 | 0.016 | 0.030 | 0.016 | 0.032 | 0.012 | 0.016 | 0.031 | 0.060 |
| 1800 | 0.015 | 0.029 | 0.016 | 0.031 | 0.012 | 0.015 | 0.033 | 0.062 |

Table 6: fSNO train and test results for all 2D datasets, $N_{\text{train}} = 1800$, $N_{\text{test}} = 200$.

| equation | $E_{\text{train}}$ | $E_{\text{test}}$ | $E_{\text{train}}^{\text{up}}$ | $E_{\text{test}}^{\text{up}}$ | $R_{\text{train}}^{\text{up}}$ | $R_{\text{test}}^{\text{up}}$ |
|---|---|---|---|---|---|---|
| (23) | 0.012 | 0.013 | 0.024 | 0.031 | 0.844 | 0.821 |
| (24) | 0.017 | 0.018 | 0.023 | 0.023 | 0.795 | 0.819 |
| (25) | 0.014 | 0.017 | 0.022 | 0.028 | 0.938 | 0.929 |
| (26) | 0.013 | 0.014 | 0.027 | 0.034 | 0.844 | 0.834 |

Table 7: ChebNO train and test results for all 2D datasets, $N_{\text{train}} = 1800$, $N_{\text{test}} = 200$.

| equation | $E_{\text{train}}$ | $E_{\text{test}}$ | $E_{\text{train}}^{\text{up}}$ | $E_{\text{test}}^{\text{up}}$ | $R_{\text{train}}^{\text{up}}$ | $R_{\text{test}}^{\text{up}}$ |
|---|---|---|---|---|---|---|
| (23) | 0.013 | 0.015 | 0.028 | 0.034 | 0.858 | 0.847 |
| (24) | 0.017 | 0.018 | 0.025 | 0.025 | 0.767 | 0.796 |
| (25) | 0.015 | 0.018 | 0.024 | 0.030 | 0.941 | 0.933 |
| (26) | 0.013 | 0.014 | 0.028 | 0.034 | 0.859 | 0.831 |

Table 8: UNet train and test results for all 2D datasets, $N_{\text{train}} = 1800$, $N_{\text{test}} = 200$.

| equation | $E_{\text{train}}$ | $E_{\text{test}}$ | $E_{\text{train}}^{\text{up}}$ | $E_{\text{test}}^{\text{up}}$ | $R_{\text{train}}^{\text{up}}$ | $R_{\text{test}}^{\text{up}}$ |
|---|---|---|---|---|---|---|
| (23) | 0.012 | 0.014 | 0.023 | 0.035 | 0.872 | 0.823 |
| (24) | 0.025 | 0.024 | 0.037 | 0.036 | 0.869 | 0.883 |
| (25) | 0.070 | 0.078 | 0.128 | 0.142 | 0.824 | 0.819 |
| (26) | 0.013 | 0.014 | 0.026 | 0.035 | 0.866 | 0.820 |

Table 9: DilResNet train and test results for all 2D datasets, $N_{\text{train}} = 1800$, $N_{\text{test}} = 200$.

| equation | $E_{\text{train}}$ | $E_{\text{test}}$ | $E_{\text{train}}^{\text{up}}$ | $E_{\text{test}}^{\text{up}}$ | $R_{\text{train}}^{\text{up}}$ | $R_{\text{test}}^{\text{up}}$ |
|---|---|---|---|---|---|---|
| (23) | 0.008 | 0.009 | 0.015 | 0.018 | 0.823 | 0.771 |
| (24) | 0.010 | 0.012 | 0.016 | 0.017 | 0.807 | 0.826 |
| (25) | 0.009 | 0.011 | 0.014 | 0.019 | 0.890 | 0.881 |
| (26) | 0.009 | 0.010 | 0.015 | 0.018 | 0.818 | 0.803 |

Table 10: Results for distinct equations averaged with respect to architectures, $N_{\text{train}} = 1200$.

| equation | $E_{\text{train}}$ | $E_{\text{test}}$ | $E_{\text{train}}^{\text{ub}}$ | $E_{\text{test}}^{\text{ub}}$ | $R_{\text{train}}^{\text{ub}}$ | $R_{\text{test}}^{\text{ub}}$ | $\widetilde{E}_{\text{train}}^{\text{ub}}$ | $\widetilde{E}_{\text{test}}^{\text{ub}}$ | $\widetilde{R}_{\text{train}}^{\text{ub}}$ | $\widetilde{R}_{\text{test}}^{\text{ub}}$ |
|---|---|---|---|---|---|---|---|---|---|---|
| (23) | 0.009 | 0.012 | 0.017 | 0.023 | 0.861 | 0.859 | 0.117 | 0.125 | 0.840 | 0.831 |
| (24) | 0.015 | 0.017 | 0.031 | 0.034 | 0.980 | 0.986 | 0.103 | 0.107 | 0.832 | 0.841 |
| (25) | 0.015 | 0.020 | 0.027 | 0.034 | 0.977 | 0.984 | 0.113 | 0.138 | 0.752 | 0.794 |
| (26) | 0.010 | 0.013 | 0.017 | 0.021 | 0.861 | 0.877 | 0.111 | 0.116 | 0.789 | 0.786 |

Table 11: Results for distinct architectures averaged with respect to equations, $N_{\text{train}} = 1200$.

| Architecture | $E_{\text{train}}$ | $E_{\text{test}}$ | $E_{\text{train}}^{\text{ub}}$ | $E_{\text{test}}^{\text{ub}}$ | $R_{\text{train}}^{\text{ub}}$ | $R_{\text{test}}^{\text{ub}}$ | $\widetilde{E}_{\text{train}}^{\text{ub}}$ | $\widetilde{E}_{\text{test}}^{\text{ub}}$ | $\widetilde{R}_{\text{train}}^{\text{ub}}$ | $\widetilde{R}_{\text{test}}^{\text{ub}}$ |
|---|---|---|---|---|---|---|---|---|---|---|
| ChebNO | 0.012 | 0.014 | 0.021 | 0.024 | 0.916 | 0.922 | 0.092 | 0.097 | 0.911 | 0.906 |
| DilResNet | 0.013 | 0.016 | 0.023 | 0.035 | 0.939 | 0.935 | 0.072 | 0.077 | 0.779 | 0.838 |
| FNO | 0.009 | 0.016 | 0.017 | 0.026 | 0.911 | 0.935 | 0.133 | 0.168 | 0.818 | 0.838 |
| MLP | 0.020 | 0.020 | 0.036 | 0.034 | 0.922 | 0.916 | 0.166 | 0.167 | 0.709 | 0.693 |
| UNet | 0.009 | 0.013 | 0.018 | 0.023 | 0.923 | 0.929 | 0.090 | 0.096 | 0.783 | 0.777 |
| fSNO | 0.011 | 0.013 | 0.022 | 0.025 | 0.907 | 0.921 | 0.114 | 0.124 | 0.820 | 0.827 |

Table 12: Results for distinct $N_{\text{train}}$ averaged with respect to the network type and equation.

| $N_{\text{train}}$ | $E_{\text{train}}$ | $E_{\text{test}}$ | $E_{\text{train}}^{\text{ub}}$ | $E_{\text{test}}^{\text{ub}}$ | $R_{\text{train}}^{\text{ub}}$ | $R_{\text{test}}^{\text{ub}}$ | $\widetilde{E}_{\text{train}}^{\text{ub}}$ | $\widetilde{E}_{\text{test}}^{\text{ub}}$ | $\widetilde{R}_{\text{train}}^{\text{ub}}$ | $\widetilde{R}_{\text{test}}^{\text{ub}}$ |
|---|---|---|---|---|---|---|---|---|---|---|
| 200 | 0.039 | 0.052 | 0.113 | 0.190 | 0.916 | 0.925 | 0.263 | 0.322 | 0.768 | 0.819 |
| 400 | 0.023 | 0.030 | 0.049 | 0.073 | 0.917 | 0.930 | 0.177 | 0.206 | 0.795 | 0.813 |
| 600 | 0.018 | 0.023 | 0.033 | 0.047 | 0.915 | 0.925 | 0.145 | 0.165 | 0.794 | 0.809 |
| 800 | 0.015 | 0.019 | 0.027 | 0.035 | 0.915 | 0.925 | 0.135 | 0.149 | 0.804 | 0.818 |
| 1000 | 0.013 | 0.017 | 0.024 | 0.031 | 0.914 | 0.924 | 0.116 | 0.129 | 0.795 | 0.809 |
| 1200 | 0.012 | 0.015 | 0.023 | 0.028 | 0.920 | 0.926 | 0.111 | 0.122 | 0.803 | 0.813 |

Table 13: Equation (23), different resolutions, results are averaged with respect to architecture type.

| $J$ | $E_{\text{train}}$ | $E_{\text{test}}$ | $E_{\text{train}}^{\text{ub}}$ | $E_{\text{test}}^{\text{ub}}$ | $\widetilde{E}_{\text{train}}^{\text{ub}}$ | $\widetilde{E}_{\text{test}}^{\text{ub}}$ | $\widetilde{E}_{\text{train}}^{\text{ub}}/E_{\text{train}}^{\text{ub}}$ | $\widetilde{E}_{\text{test}}^{\text{ub}}/E_{\text{test}}^{\text{ub}}$ |
|---|---|---|---|---|---|---|---|---|
| 5 | 0.009 | 0.012 | 0.017 | 0.023 | 0.117 | 0.125 | 7.019 | 5.448 |
| 6 | 0.012 | 0.015 | 0.024 | 0.027 | 0.127 | 0.135 | 5.364 | 5.020 |
| 7 | 0.020 | 0.022 | 0.060 | 0.058 | 0.183 | 0.183 | 3.043 | 3.174 |

Table 14: Ratios $\frac{\text{upper bound obtained by a posteriori neural solver}}{\text{upper bound obtained by optimization}}$, $N_{\text{train}} = 1200$.

| architecture \ equation | train set | | | | test set | | | |
|---|---|---|---|---|---|---|---|---|
| | (23) | (24) | (25) | (26) | (23) | (24) | (25) | (26) |
| ChebNO | 4.5 | 3.8 | 3.0 | 5.2 | 5.4 | 4.0 | 3.3 | 5.9 |
| DilResNet | 2.7 | 1.5 | 2.2 | 3.3 | 4.0 | 2.4 | 2.9 | 3.6 |
| FNO | 6.5 | 5.9 | 6.6 | 7.3 | 10.1 | 5.6 | 6.7 | 10.0 |
| MLP | 8.6 | 1.6 | 5.7 | 5.8 | 8.9 | 1.6 | 5.0 | 6.0 |
| UNet | 4.6 | 3.3 | 3.6 | 5.8 | 6.9 | 3.4 | 3.4 | 8.4 |
| fSNO | 5.2 | 4.8 | 5.0 | 5.5 | 5.9 | 4.9 | 4.8 | 6.1 |

Table 15: Results for fSNO for different $\lambda$ for 2D datasets, $N_{\text{train}} = 1200$, $N_{\text{test}} = 800$

| equation | $\lambda = 0.01$ | | $\lambda = 0.1$ | | $\lambda = 1$ | | $\lambda = 10$ | | $\lambda = 100$ | |
|---|---|---|---|---|---|---|---|---|---|---|
| | $E_{\text{train}}$ | $E_{\text{test}}$ | $E_{\text{train}}$ | $E_{\text{test}}$ | $E_{\text{train}}$ | $E_{\text{test}}$ | $E_{\text{train}}$ | $E_{\text{test}}$ | $E_{\text{train}}$ | $E_{\text{test}}$ |
| (23) | 0.026 | 0.027 | 0.016 | 0.018 | 0.014 | 0.016 | 0.017 | 0.019 | 0.104 | 0.102 |
| (24) | 0.028 | 0.028 | 0.02 | 0.02 | 0.019 | 0.019 | 0.021 | 0.021 | 0.03 | 0.029 |
| (25) | 0.021 | 0.023 | 0.015 | 0.017 | 0.016 | 0.018 | 0.017 | 0.019 | 0.071 | 0.070 |
| (26) | 0.032 | 0.033 | 0.016 | 0.019 | 0.014 | 0.017 | 0.017 | 0.019 | 0.105 | 0.105 |

Table 16: Results for ChebNO for different $\lambda$ for 2D datasets, $N_{\text{train}} = 1200$, $N_{\text{test}} = 800$

| equation | $\lambda = 0.01$ | | $\lambda = 0.1$ | | $\lambda = 1$ | | $\lambda = 10$ | | $\lambda = 100$ | |
| | $E_{\text{train}}$ | $E_{\text{test}}$ | $E_{\text{train}}$ | $E_{\text{test}}$ | $E_{\text{train}}$ | $E_{\text{test}}$ | $E_{\text{train}}$ | $E_{\text{test}}$ | $E_{\text{train}}$ | $E_{\text{test}}$ |
|---|---|---|---|---|---|---|---|---|---|---|
| (23) | 0.026 | 0.028 | 0.016 | 0.018 | 0.015 | 0.017 | 0.016 | 0.018 | 0.064 | 0.062 |
| (24) | 0.032 | 0.031 | 0.021 | 0.021 | 0.02 | 0.02 | 0.021 | 0.021 | 0.028 | 0.027 |
| (25) | 0.024 | 0.026 | 0.016 | 0.018 | 0.016 | 0.018 | 0.02 | 0.021 | 0.055 | 0.054 |
| (26) | 0.03 | 0.031 | 0.017 | 0.019 | 0.016 | 0.017 | 0.019 | 0.021 | 0.025 | 0.026 |

Table 17: Results for UNet for different $\lambda$ for 2D datasets, $N_{\text{train}} = 1200$, $N_{\text{test}} = 800$

| equation | $\lambda = 0.01$ | | $\lambda = 0.1$ | | $\lambda = 1$ | | $\lambda = 10$ | | $\lambda = 100$ | |
| | $E_{\text{train}}$ | $E_{\text{test}}$ | $E_{\text{train}}$ | $E_{\text{test}}$ | $E_{\text{train}}$ | $E_{\text{test}}$ | $E_{\text{train}}$ | $E_{\text{test}}$ | $E_{\text{train}}$ | $E_{\text{test}}$ |
|---|---|---|---|---|---|---|---|---|---|---|
| (23) | 0.02 | 0.021 | 0.013 | 0.015 | 0.015 | 0.015 | 0.021 | 0.022 | 0.104 | 0.102 |
| (24) | 0.04 | 0.04 | 0.028 | 0.027 | 0.019 | 0.019 | 0.091 | 0.09 | 0.11 | 0.109 |
| (25) | 0.024 | 0.024 | 0.071 | 0.07 | 0.071 | 0.07 | 0.071 | 0.07 | 0.071 | 0.07 |
| (26) | 0.016 | 0.017 | 0.016 | 0.018 | 0.012 | 0.015 | 0.032 | 0.035 | 0.105 | 0.105 |

Table 18: Results for DilResNet for different $\lambda$ for 2D datasets, $N_{\text{train}} = 1200$, $N_{\text{test}} = 800$

| equation | $\lambda = 0.01$ | | $\lambda = 0.1$ | | $\lambda = 1$ | | $\lambda = 10$ | | $\lambda = 100$ | |
| | $E_{\text{train}}$ | $E_{\text{test}}$ | $E_{\text{train}}$ | $E_{\text{test}}$ | $E_{\text{train}}$ | $E_{\text{test}}$ | $E_{\text{train}}$ | $E_{\text{test}}$ | $E_{\text{train}}$ | $E_{\text{test}}$ |
|---|---|---|---|---|---|---|---|---|---|---|
| (23) | 0.013 | 0.013 | 0.01 | 0.01 | 0.009 | 0.01 | 0.009 | 0.01 | 0.017 | 0.017 |
| (24) | 0.021 | 0.02 | 0.018 | 0.018 | 0.018 | 0.018 | 0.018 | 0.018 | 0.027 | 0.026 |
| (25) | 0.013 | 0.014 | 0.012 | 0.012 | 0.011 | 0.012 | 0.012 | 0.012 | 0.048 | 0.047 |
| (26) | 0.013 | 0.014 | 0.009 | 0.01 | 0.01 | 0.01 | 0.01 | 0.011 | 0.026 | 0.026 |

