# OpenReview forum: "Neural functional a posteriori error estimates"
_ICLR.cc/2024/Conference — Submitted to ICLR 2024_

### Official Review · Reviewer_h1aP · 2023-10-27

**Soundness:** 3 good
**Presentation:** 1 poor
**Contribution:** 3 good
**Rating:** 6
**Confidence:** 1

**Summary:**

This paper brings "functional a posteriori error analysis" from the mathematics community to machine learning. Similar to Hillebrecht et al and many others, the goal is to obtain error certificates, i.e, some formal statement on the upper bound of the error produced by physics-informed neural networks. In functional a posteriori error analysis, the upper bound of the error depends on approximate solution, data, and additional fields that can potentially tighten the bound. This work builds on this direction and proposes a loss function Astral. Experiments show that in an unsupervised setting, Astral outperforms one baseline (Li et al).

**Strengths:**

I note that I am not an expert in the related domain of the paper. Hence, I requested area chairs for additional reviewers. My score reflects that I am ignorant in this field and do not want to reject papers.

There are many papers on certifiable machine learning, where the goal is to analyze the error, traditionally from generalization theory like PAC-Bayes bounds. For physics-informed neural networks, I think there might be meaning in investigating practical techniques to upper bound the error.

What this paper provides is perhaps the idea of bringing "functional a posteriori error analysis", which can be interesting to certain parts of the community.

**Weaknesses:**

On the other hand, I would have preferred if the presentation of the paper was more kind to readers. For example:

The introduction provides minimum information about astra -- only that it is a loss function with certain benefits like more robustness over residuals and variational losses. In my view, there should be a high-level description of (a) what motivates this specific loss function, and (b) what exactly is this loss function. There should also be reasoning behind it at a high level. It is difficult to grasp the concept and also get interested otherwise. I think the paper can also be more kind to the readers by defining certain terminology before using them, e.g., majorants, posterior error, priori error, etc.

I found section 2.3 to be difficult to parse. The paper states several equations without explaining them in sufficient depth. For example, in equation (9), I did not understand where the definition of astral is from. The paper states "It is possible to derive" but at least in the appendix, these derivations should be shown. Equation 12 is also similar. I could not understand the derivation from equation 9 to equation 12. There should also be sufficient reasoning "why", for example, equation 12 is a good measure of the predicted solution.

I hope other reviewers can comment more in-depth about technical part of the paper and quality of the experiments.

**Questions:**

My questions for clarification are embedded in the comments above.

---

> ### Author Response · Authors · 2023-11-19
>
> We appreciate the review and thank the reviewer for providing comments on the article.
>
> We understand that in parts the presented work may be hard to follow. This is, however, subjective, since we can see that pN9X, and SLC5 rank presentation as good and excellent, CFkx ranks it as fair and finally, the present reviewer h1aP ranks it as poor. Given that we have a full spectrum of opinions.
>
> > The introduction provides minimum information about astra -- only that it is a loss function with certain benefits like more robustness over residuals and variational losses. In my view, there should be a high-level description of (a) what motivates this specific loss function, and (b) what exactly is this loss function. There should also be reasoning behind it at a high level. It is difficult to grasp the concept and also get interested otherwise.
>
> There is a part of the paper that provides high-level details on the Astral loss. We tried to convey this information in 2.1, where the rationale behind using the functional error estimate as a loss function is given (certain favorable properties are listed).
>
> > I think the paper can also be more kind to the readers by defining certain terminology before using them, e.g., majorants, posterior error, priori error, etc.
>
> We agree that these terms are not properly defined. We will make sure to provide brief comments when appropriate. More specifically, we will comment on the terms majorant and minorant right after equation (2) where they first appeared. Similarly, we will comment on a priori and a posteriori errors in the introduction where these terms appeared first.
>
> > I found section 2.3 to be difficult to parse. The paper states several equations without explaining them in sufficient depth. For example, in equation (9), I did not understand where the definition of astral is from. The paper states "It is possible to derive" but at least in the appendix, these derivations should be shown. Equation 12 is also similar. I could not understand the derivation from equation 9 to equation 12
>
> We agree that the presentation is sketchy here. The main reason why is the limit on the length of the paper. We provided references in the text sufficient for the reproduction of equations but we agree that it is better to make paper as self-sufficient as possible. Shortly we will provide a revised version of the manuscript with two additional appendices: the first one with the derivation of upper bound (9), and the second one with the explanation of how to arrive at (12) from (9).
>
> > There should also be sufficient reasoning "why", for example, equation 12 is a good measure of the predicted solution.
>
> We agree that this is an important question and thank the reviewer for bringing this up. Indeed, the manuscript does not explain properly why the energy norm (and its upper bound) is a good measure of accuracy. In some sense this material is standard, but we will review it properly. For details we refer to the answer https://openreview.net/forum?id=z62Xc88jgF&noteId=SQ0cDlqod2, where we address this question in some detail.

---

### Official Review · Reviewer_SLC5 · 2023-10-28

**Soundness:** 3 good
**Presentation:** 3 good
**Contribution:** 3 good
**Rating:** 6
**Confidence:** 3

**Summary:**

This paper proposes a new loss function for supervised and physics-informed training of neural networks and operators that incorporates a posteriori error estimate. The trained model can guarantee the approximation error as this is done by the new loss function aiming at minimizing the theoretical posterior error estimate. The later has been established for a number of PDEs.

**Strengths:**

1.  It is a novel idea to adopt the theoretical  functional a posteriori error estimates for the learning objective. The functional a posteriori error estimate was initially established in the conventional finite element method analysis for PDE.

2. The entire framework has been clearly described (at least I can follow the main stream of the paper, although I am not the expert in this particular area).

3. The paper is well motivated with the goal to mitigate neural PDE solvers' inability guarantee good accuracy in practice.

**Weaknesses:**

Although this is a viable approach in a guaranteed way to produce reliable neural PDE solvers, the application could be limited, e.g., in the case for the problems where there is no theoretical posteriori estimates available.

**Questions:**

Not sure why the full GitHub repository is not made available for review.

---

> ### Author Response · Authors · 2023-11-19
>
> We want to thank the reviewer for taking part in the discussion of the manuscript.
>
> We agree that the approach is limited to the PDEs where the functional upper bound can be derived, and this is the main downside. However, for many practically relevant equations, the functional error estimate exists. A non-exhaustive list of these equations is given at the end of 2.3 of the present manuscript.
>
> Here, we would like to mention that error majorants are available for the Maxwell equation and a wide class of variational problems with convex functions.
>
> Besides that, we provide additional examples of how to apply functional error estimate: $D=2$ elliptic equation in the L-shaped, $D=1+1$ convection-diffusion equation. More details are available in https://openreview.net/forum?id=z62Xc88jgF&noteId=Wy4TDErd0T.
>
> As for the code, we decided not to share the GitHub repository because direct sharing is prohibited by rules. On the other hand, anonymization of the repository is tedious. The Google Colab notebooks that we share are anonymous and can be immediately used to reproduce the results since they run in the browser. We will make the code available after the review stage.

---

> > ### Comment · Reviewer_SLC5 · 2023-11-22
> >
> > Thanks for addressing my comments.
> >
> > I still think releasing code is helpful in review. To protect your identity, you have a lot of ways to do so, e.g., https://anonymous.4open.science/ or as supplementary to your paper submission on openreivew.

---

> > > ### Author Response · Authors · 2023-11-22
> > >
> > > We would like to thank the reviewer for the suggestion. Using the proposed approach, we created a link to the GitHub repository that contains the code, including all notebooks, architectures, transformations, and scripts used for training https://anonymous.4open.science/r/UQNO-23BC. We will also attach the same link to the revision of the paper that will appear shortly.

---

> ### Comment · Reviewer_SLC5 · 2023-11-22
>
> Thanks for releasing the code.   I am revising my rating up.

---

### Official Review · Reviewer_pN9X · 2023-11-01

**Soundness:** 3 good
**Presentation:** 4 excellent
**Contribution:** 3 good
**Rating:** 8
**Confidence:** 3

**Summary:**

The paper proposes a method for training neural networks that provide solutions to PDEs (including PINNs, Neural Operator networks, and vanilla surrogate models). The idea is to upper bound the error of the predicted solutions and incorporate this upper bound into the training loss of the neural network. Constructing the the upper bound on the solution error is non-trivial and requires expert knowledge, but this provides a way of incorporating domain knowledge into the model.

**Strengths:**

- The paper is well-written and easy to follow. Figures are clear and informative.
- The paper addresses a method for computing error bounds on solutions to PDEs, a challenge of high interest to the ML for physics community.
- Experiments support the claims.

**Weaknesses:**

- It seems like this approach will have limited applicability. The first limitation is that the U function must be specified by the practitioner, but this is addressed in the paper, seems reasonable, and is a way to incorporate domain knowledge. The bigger issue is that in most scenarios, I imagine that predicting the error certificates is at least as difficult (and usually more difficult) than predicting the solution.

**Questions:**

- My intuition is that the error certificate w_U can be thought of as an *explanation* of how the solution emerges. For example, the certificate could explain the predictions of a day-ahead weather forecast model by providing the evolution of the weather variables over the intermediate time steps. When a good certificate can be generated, the practitioner can feel confident that the solution is correct (with hard error bounds!), but otherwise the error bounds will be loose and the practitioner will not have much confidence in the solution. This all seems desirable, but I wonder if there exists a large set of problems for which the surrogate model can produce good solutions but not produce good certificates?

There are some minor typos throughout:
- Figure 1: should approximate solution u by u^tilde?
- Section 2.1 "deep learning is to"

---

> ### Public Comment · ~Shuhao_Cao1 · 2023-11-12
>
> "Constructing the the upper bound on the solution error is non-trivial. "
> - No. The upper bound is trivially obtained for elliptic problems if one uses functional-type a posteriori error estimation. For Poisson problems, inserting the flux variable $\\|\nabla u  - \nabla u_h\\|^2$ and integration by parts you are good to go. The difficult part is to the lower bound ***locally***, in the sense that the local error can be also bounded below by the error estimator (local version). Check Repin's book chapter 2.2.2.2 for example, and also chapter 5 about the local efficiency. For residual-based a posteriori error estimation approaches (dated back to 60s which the authors cited none), the difficulty reverses (local efficiency is trivial but the upper bound needs the discretization's stability and a quasi-interpolant).
>
> "My intuition is that the error certificate $w_U$ can be thought of as an explanation of how the solution emerges"
> - $\\|u - u_h\\|_V$ is not computable, because $u$ is unknown (unless in a toy model). If $V$ is Hilbert, then this error is a functional in $V'$, $w_U$ can be viewed as an approximation to the Riesz representation of this functional in a computable norm $L^2$.

---

> ### Author Response · Authors · 2023-11-19
>
> We would like to thank the reviewer for the time spent reading the paper and for the feedback.
>
> Typos:
>
> - Indeed, the approximate solution should be $\widetilde{u}$. We will fix that in the revision.
> - “deep learning is to” --> “deep learning to”
>
> As for the question. Indeed, the solution may be good be the certificate produced by the surrogate model is not good, i.e., they lead to an upper bound that is much larger than the energy norm. In our experiments, however, we did not observe this behavior when input data is in distribution. Out of distribution, this may happen, but the situation is hard to analyze. In this unfortunate case when certificates are useless, the practitioner has two choices:
>
> 1. Recompute the solution with a more reliable method.
> 2. Optimize the loss with a fixed solution to find better certificates.
>
> So the certificates are useful only in the situation when the predicted upper bound is smaller than the desired tolerance. The situation in machine learning and statistics when the method provides a measure of certainty for the obtained result (e.g., vote counting, document sorting). When the measure is good enough, no intervention is needed. When the measure is bad, the sample is processed with a more reliable method (e.g., by a human being).
>
> > Constructing the the upper bound on the solution error is non-trivial and requires expert knowledge, but this provides a way of incorporating domain knowledge into the model.
>
> Indeed, we agree that the construction of the upper bound is not straightforward and requires expert knowledge. We recognize this as a weakness of the approach and provide some discussion in these lines in https://openreview.net/forum?id=z62Xc88jgF&noteId=L8Rln3shGY. This discussion includes the list of equations where the upper bound is known and the additional example of the convection-diffusion equation that we introduce to showcase that not only elliptic equations can be treated with functional a posteriori error estimate (see https://openreview.net/forum?id=z62Xc88jgF&noteId=Wy4TDErd0T).

---

> ### Comment · Reviewer_pN9X · 2023-11-23
> **Update after reading comments from other reviewers and the responses**
>
> I have one of the most positive reviews for this paper because I think it is well-written and brings new ideas to the ML community. I admit that it is out of my area of expertise and the other reviewers likely have more knowledge of this area than me, but I think the authors have responded well to the criticisms and comments, so I see no reason to change my positive review.
>
> In particular, the public comments from an expert in this area were critical of the paper's novelty. (https://openreview.net/forum?id=z62Xc88jgF&noteId=1MwG3D88ca) The responses from the authors seem thorough and make sense to me, but I admit I do not have the depth of expertise to be confident in my judgement.

---

### Official Review · Reviewer_CFkx · 2023-11-08

**Soundness:** 2 fair
**Presentation:** 1 poor
**Contribution:** 2 fair
**Rating:** 3
**Confidence:** 3

**Summary:**

This paper propose a loss function for training PINNs with theoretically guaranteed error bounds. Experiments are done for verifying their claims.

**Strengths:**

1. By the theory from functional a posteriori error analysis, using Astral loss gives theoretical guarantee that the output of neural network is the exact solution in the sense that their distance is zero in some function space.

2. Astral loss can be computed explicitly for common PDE problems. The authors use elliptic equation as an example.

3. Some experiments are done to compare Astral loss with the commonly used residual loss using different models/equations.

**Weaknesses:**

1. Although the use of this type of loss in this setting might be new, this work does not prove any new theoretical results.

2. That being said, experiment is a very important component in this paper, however, I find the evaluation metric of the solution very interesting. More specifically, let $u$ be the output of neural networks and $u^*$ be the exact solution. The test error is usually computed using relative $L^2$ norm (See for example [1][2]), i.e.
$$|| u - u^*||_2^2 / ||u^*||_2^2 = \int|u - u^*|^2dx / \int |u^*|^2 dx.$$
However, in Figure 4, when evaluating solutions, the mean error is computed using equation (15), the energy norm.

(i). why not using the relative $L^2$ norm? How does Astral loss perform if the evaluation is done in $L^2$?

(ii). The a posteriori error bound is in the energy norm, i.e.
$$L(u, w_L) \leq |||u-u^*||| \leq U(u, w_U).$$
so I would naturally expect Astral loss to achieve fairly small error in this energy norm, but this does not necessarily imply the solution is "better". Equations can be solved in different spaces. In fact, I think the space $L^2$ is more commonly used when people study existence and uniqueness of PDE solutions.

(iii). There could be a relation between the energy norm and $L^2$ norm. More explanation is needed for the specific choice of the evaluation metric since it differs from the previous literature.


[1] Li et al., Physics-Informed Neural Operator for Learning Partial Differential Equations

[2] Wang et al., An Expert's Guide to Training Physics-informed Neural Networks

**Questions:**

See weaknesses.

---

> ### Author Response · Authors · 2023-11-19
>
> We would like to thank the reviewer for bringing up an important question. We agree that the relation between errors computed in energy and the $L_2$ norm is an important topic not covered in the article. There are two parts to consider: the theoretical one — how the losses are related, and the practical one — what are typical magnitudes of errors in each norm when networks are trained with Astral and residual losses.
>
> The theoretical part is as follows. Suppose the error is $e$, the diffusion coefficient is $a$, and the reaction term is $b^2$. In this case, we can find
>
> $$|||e|||^2 = \int_{\Gamma} dx \sum_{ij} \frac{\partial e}{\partial x_{i}}\frac{\partial e}{\partial x_{j}}a_{ij} + ||be||^2 \geq \lambda_{\min}^2 ||e||^2 + ||be^2|| \geq \lambda_{\min}^2 ||e||^2 + \inf_{x} b(x)^2||e^2||,$$
>
> where $\lambda_{\min}$ is a minimal eigenvalue of the elliptic problem $-\frac{\partial}{\partial x_{i}} a_{ij} \frac{\partial}{\partial x_{j}}u(x) = \lambda u(x)$ defined on the same domain and with the same boundary conditions as the original elliptic problem. From the expression above, we obtain the bound
>
> $$||e||\_{2}^{2} \\leq \\frac{1}{\\lambda_{\\min}^2 + \\inf_{x} \\left(b(x)\right)^2} |||e|||.$$
>
> This bound can be simplified and made more practical (e.g., by removing minimal eigenvalue that may be hard to compute) by various means but the bound above is already sufficient to claim that when error is sufficiently small in the energy norm, this is also the case for the standard $L_2$ norm.
>
> To answer the practical part, we conducted the set of additional experiments described in the reply https://openreview.net/forum?id=z62Xc88jgF&noteId=Wy4TDErd0T. There, we compared relative errors and energy norms for two additional test cases ($D=2$ elliptic equation in L-shaped domain, $D=1+1$ convection-diffusion equation) for Astral and residual losses.
>
> We would like to also add that, indeed, solutions of PDE can be considered in various spaces. Most typically, however, not a space $L_2$ is used but Sobolev spaces to properly define weak solutions (see [E] Part II). Errors in natural Sobolev norms are sometimes used in deep learning literature. For example, a recent paper with FNO extension uses this norm [K]. Other metrics besides relative $L_2$ norms are also used, e.g., $L_1$ norm divided by $L_{\infty}$ [B], relative RMSE of energy spectrum (turbulence) [C], etc
>
> The main reason we use energy norms is because it is a natural norm for elliptic problems. Namely, the energy norm will weigh the error according to the importance of spaces spanned by eigenvalues of the elliptic operator in question (eigenspaces with larger eigenvalues receive more weight). The energy norm is a de facto standard for elliptic PDEs. For example, FEM error estimate is done with error energy norm [SB, for example, 1.5.2 and 1.5.3]; the convergence theory for Conjugate Gradient method (Krylov subspace method used to solve a linear system with symmetric positive-definite matrices including the ones originated from elliptic equations) is naturally formulated with energy norm [S].
>
> Finally, we would like to point out that we do not claim that the solution obtained with Astral loss is better. Our main claim is that Astral loss provides a natural way to construct an error majorant that provides a tight bound on the error. Note that the residual does not have this property. It is easy to see that the residual can be arbitrarily bad for the solution that is arbitrarily close to the exact one. Namely, we can consider a small high-frequency perturbation $\\delta(x) = \\epsilon\\sin(2\\pi L x)$ to the exact solution. This perturbation will introduce an error of the order $\epsilon$ and residual of order $\left(2 \\pi L\right)^2 \epsilon$. Now one can take $L$ large enough to have an arbitrarily large residual.
>
> We hope that this discussion clarifies the relation between $L_2$ and energy norms and explains our choice of metric. We will incorporate this material in the corrected version of the paper that will appear shortly.
>
> [E] — Evans LC. Partial differential equations. American Mathematical Society; 2022 Mar 22.
>
> [K] — https://arxiv.org/abs/2310.00120v1
>
> [B] — https://arxiv.org/abs/2202.03376
>
> [C]— https://arxiv.org/abs/2112.01527
>
> [SB] — Szabó B, Babuška I. Finite Element Analysis: Method, Verification and Validation.
>
> [S] — Shewchuk JR. An introduction to the conjugate gradient method without the agonizing pain.

---

### Public Comment · ~Shuhao_Cao1 · 2023-11-12

If the flux variable is obtained by minimizing the "astral loss" (5), and more specifically the one for elliptic problems on page 5, then this is exactly the same method with [CCLX], equation (2.6) in the arXiv version, $\boldsymbol{\tau}$ is $y$ here. It is named "FOSLS" functional therein, check (3.6).

[CCLX]: Z Cai, J Chen, M Liu, X Liu - J. Comput. Phys 2020, arXiv:1911.02109

There are lots of missing references here on the newer development since Repin. Too bad it seems that none of the reviewers have ever worked in the area of *a posteriori* error estimation for traditional discretization methods (FEM or FVM or dG).

There are many errors, to list a few.
-  The approach in Gupta et al. (2021) is not transformer-based.
- $a_{ij}\geq c>0$ is not the sufficient condition that equation (6) is elliptic.
- PDE is usually defined on open domains, not closed ones.
- It is "Cauchy–Schwarz", not "Cauchy–Schwartz".
- equation (10) is wrong.

None of the standard benchmark problems for a posterior error estimation is used, especially solutions with only $H^{1+\epsilon}$ regularity ($0<\epsilon\leq 1/2$), to name a few: Kellogg's intersecting interfaces for $a_{ij}(\cdot)$, L-shaped corner or Fichera cube, domain with a crack, etc.

---

> ### Author Response · Authors · 2023-11-19
> **comment part 1**
>
> We would like to thank Shuhao Cao for providing public comment with suggestions. We will treat this comment as an additional review given the background and keen interest of the author.
>
> There are several issues raised by Shuhao Cao:
>
> 1. Paper [1] is claimed to contain similar results.
>
>     We respectfully disagree that this contribution contains Astral loss defined in the present publication. To help other reviewers make up their minds, we provide a list of differences between our work and [1]:
>
>     1. In [1] authors consider a neural network that takes the set of points $\mathbb{R}^{n}$ as an input and produces $\mathbb{R}^{n}$ solution values. We consider physics-informed neural networks and neural operators, i.e., our formulation is continuous.
>     2. Equation (3.6) pointed out by Shuhao Cao is a discrete version of the loss function that can not be directly compared with (5) — continuous Astral loss defined for abstract PDE with known functional error estimate. It is more appropriate to compare FOSL functional (2.6) with concrete Astral loss for the elliptic equation given in (9). Undoubtedly, they are related since both of them reformulate elliptic problems in terms of fluxes (usual practice for a posteriori estimate) but there are a few significant differences:
>         1. The claim that they are the same is exaggerated. For example, the terms with fluxes are different. Besides, Astral loss contains additional parameter $\beta$ that enters the optimization problem and the upper bound on Friedrichs's constant $C$.
>         2. The FOSL loss is a posteriori error estimate and Astral is an error majorant. This means we know Astral always bound error from above — the property that we highlight through the text of the paper. For the FOSL loss, this is not the case.
>     3. In paper [1] the results are given only for ordinary differential equations (boundary value problems) in a non-parametric setting. In our paper, we consider parametric and non-parametric PDEs.
>
> 2. Missing references.
>
>     More specifically Shuhao Cao claims
>
>     > There are lots of missing references here on the newer development since Repin. Too bad it seems that none of the reviewers have ever worked in the area of *a posteriori* error estimation for traditional discretization methods (FEM or FVM or dG).
>
>     We mostly reference Repin since he provided pioneering contributions to the field of a posteriori functional error estimate. We acknowledge that the literature on the a posteriori error estimate for classical discretization methods is vast and reach. However, we see no need to introduce additional references since they are largely irrelevant to our work. The functional error estimate for the elliptic equation derived by Repin remains correct (since it is supported with formal mathematical proof) even though there is “newer development” by other researchers.
>
>     We also want to highlight that the functional a posteriori error estimate is markedly different from other approaches since it does not rely on discretization and smoothness properties. This is the main reason it is uniquely suitable for deep learning applications. It is very unfortunately this approach is not widely known by deep learning practitioners. We hope that our paper will improve the situation.
>
> [1] — https://arxiv.org/abs/1911.02109

---

> ### Author Response · Authors · 2023-11-19
> **comment part 2**
>
> 3. Errors.
>
>     Without a doubt, the manuscript contains some portion of typos. We would like to thank Shuhao Cao for spotting a few. We will cover all of them one by one:
>
>     > The approach in Gupta et al. (2021) is not transformer-based.
>
>     It is not stated in our paper that the approach in Gupta is transformer-based. The whole sentence reads “This interest led to the construction of a plethora of novel efficient architectures including Fourier Neural Operator (FNO) Li et al. (2020), Deep Operator Network (DeelONet) Lu et al. (2019), **wavelet-** and transformer-based approaches Gupta et al. (2021), Hao et al. (2023).” This means Gupta et al. (2021) provide a wavelet-based approach (WNO) and Hao et al. (2023) — a transformer based.
>
>     > $a_{ij}\geq c > 0$ is not the sufficient condition that equation (6) is elliptic.
>
>     This is a classic linear algebra shortcut for comparing positive-definite matrices. So what is written in plain English is “$a$ is a positive definite matrix with eigenvalues uniformly bound from below by constant $c$ that is greater than zero”. One can find that we are aware of how to properly define elliptic PDE by, for example, consulting the appendix. In equation (17) for one of the test problems, we generated positive definite matrices with uniformly bound eigenvalues using Cholesky decomposition.
>
>     > PDE is usually defined on open domains, not closed ones.
>
>     We did not define PDE on the closed domain. Part $x\in\Gamma = [0, 1]^{D}$ is a definition of the domain itself.
>
>     > It is "Cauchy–Schwarz", not "Cauchy–Schwartz"
>
>     We will properly spell Schwarz in the revision.
>
>     > equation (10) is wrong
>
>     There is a minor typo in the equation. It read $\\sqrt{2\left(J(\widetilde{u}) - J(\widetilde{u} + w)\right)} \leq |||\widetilde{u} - u_{\text{exact}}|||^2$ and it should be $\\sqrt{2\left(J(\widetilde{u}) - J(\widetilde{u} + w)\right)} \leq |||\widetilde{u} - u_{\text{exact}}|||$. This equation provides a lower bound and is not used directly to obtain any results of the paper. This equation will likely be removed in the next revision to make space for other explanations.
>
>     In summary, we would like to thank the reviewer for vigilance. In general, typos are very minor and unimportant. Still, it is good to spot as much of them as possible. We will also make careful proofreading at the later stages to make sure to remove the rest of the typos.
>
> 4. Benchmarks.
>
>     We have two comments on the benchmarks proposed by Shuhao Cao:
>
>     1. We performed a few additional tests: an elliptic equation with corner singularity (L-shape domain) and a posteriori functional error estimate for the convection-diffusion equation. Please, see https://openreview.net/forum?id=z62Xc88jgF&noteId=Wy4TDErd0T for more details.
>     2. We do not find the standard FEM a posteriori error benchmarks helpful. There are two reasons for that:
>         1. Neural PDE solvers and classical methods differ significantly.
>
>             In classical methods, one can directly control the degree of approximation by increasing the number of degrees of freedom (e.g., the number of elements or points in physical space). Because of that classical benchmark aims to show how the situation changes for less smooth functions or domains with singularities. Often the focus is on the asymptotic properties (does the estimate converge) and on the adaptive sampling of grid points. All that is pointless for physics-informed neural networks and neural operators. The latter one does not show convergence when the grid is refined (see Figure 1 in [2]). The best result for the physics-informed neural network is often $10^{-2}$ relative error in $L_2$ norm with no sign of improvement for larger networks or finer grids (see [3] for SOTA results table 3; also table 3 from [4]). This absence of convergence makes classical benchmarks unsuitable. Indeed as shown in https://openreview.net/forum?id=z62Xc88jgF&noteId=Wy4TDErd0T nothing essentially new is observed for the L-shaped domain.
>
>         2. We use a functional a posteriori error estimate that was already tested for standard problems. We merely incorporate the known results into the deep learning approach. It is not clear why standard tests are warranted in this case.
>
> Overall, we would like to thank Shuhao Cao for spending time on reading the paper and making valuable suggestions.
>
> [1] — https://arxiv.org/abs/1911.02109
>
> [2] — https://arxiv.org/abs/2309.15325
>
> [3] — https://arxiv.org/abs/2308.08468
>
> [4] — https://arxiv.org/abs/2306.08827

---

### Author Response · Authors · 2023-11-19
**Additional benchmarks and results**

Following the suggestion of several reviewers we expanded the number of test cases with two problems and provided additional results that allowed us to compare the relative $L_2$ norm of the solution obtained with the Astral loss and the residual loss.

The first novel problem is the elliptic equation in the $L-$shaped domain. The equation reads

$$-\\text{div}\\,\\text{grad}\\,u(x, y) + \\left(b(x, y)\\right)^2 u(x, y) = f(x, y),\\,(x, y) \\in [0, 1]^2 \\backslash [0.5, 1]^2$$

with homogeneous Dirichlet conditions on the boundary. This is a classical test problem for a posteriori error estimates since it contains corner singularity.

For that problem, we generated a family of $b(x, y)$ and $f(x, y)$ using random smooth trigonometric functions and computed an accurate solution using finite difference discretization on a fine grid. This solution is used to compute the relative error.

The table below contains results for training with residual and Astral losses. Statistics is computed using $500$ equations

|loss function|relative error| energy norm|
|--|--|--|
|error majorant|$\\left(7.3 \\pm 9.8\\right)\\times 10^{-3}$ | $\\left(2.7 \\pm 4.1\\right)\times 10^{-2}$ |
|residual| $\\left(5.0 \\pm 7.2\\right)\\times 10^{-3}$ | $\\left(1.4 \\pm 1.2\\right)\times 10^{-2}$ |

Second problem we consider is a convection-diffusion equation

\begin{equation} \begin{split} &\frac{\partial u(x, t)}{\partial t} - \text{div}\\,\text{grad}\\,u(x, t) + a\cdot \text{grad}\,u(x, t) = f(x, t), \\ &u(x, 0) = \phi(x),\\ &\left.u(x, t)\right|_{x \in \partial \Gamma} = 0. \end{split} \end{equation}

Functional a posteriori error estimate is derived in https://arxiv.org/abs/1012.5089 and reads

\begin{equation} \int dx dt\\,\text{grad}\\, e(x, t) \cdot \text{grad}\\, e(x, t) + \frac{1}{2} \int dx\\,\left(e(x, T)\right)^2 \leq \int dx dt \,\left(y - \text{grad}\,v\right)\cdot \left(y - \text{grad}\,v\right) + C \int dxdt\,\left(f(x, t) - \frac{\partial v(x, t)}{\partial t} - a\cdot \text{grad}\,v(x, t) + \text{div} y(x, t)\right)^2, \end{equation}

where $e(x, t) = u(x, t) - v(x, t)$ is the error, $v(x, t)$ approximate solution and $u(x, t)$ is the exact solution and $C = \frac{1}{\pi D_x}$, where $D_{x}$ is a number of spacial dimensions. We solve the problem in $(x, t) \in [0, 1]\times[0, 0.1]$.

The table below contains results for training with residual and Astral losses for equations with randomly generated initial conditions and source terms. Statistics is computed using $500$ equations.

|loss function|relative error| energy norm|
|--|--|--|
|error majorant|$\left(8.3 \pm 8.9\right)\times 10^{-3}$ | $\left(1.3 \pm 0.6\right)\times 10^{-2}$ |
|residual|$\left(9.9 \pm 11\right)\times 10^{-3}$ | $\left(7.8 \pm 2.4\right)\times 10^{-3}$ |

To measure the quality of the upper bound for both of the new problems we computed $\frac{\\text{upper bound} - \\text{energy norm}}{\\text{energy norm}}$. The results are given in the table below.

|equation|quality of upper bound|
|--|--|
|elliptic, L-shaped|$0.84\pm0.60$|
|convection-diffusion|$0.63\pm0.18$|

Overall we observe that the relative errors and energy norms are close for residual and Astral losses. So Astral loss leads to comparable or better results and in addition to that provides a high-quality upper bound to the error in the energy norm.

Code for L-shaped domain is available in https://colab.research.google.com/drive/1tdXw6ddAsIqyLOApObUcWBqDZjknIAA2

Code for convection-diffusion problem can be found in https://colab.research.google.com/drive/1EXf5AO6kTUmu_QX_JXPc-uOE9aHgXLbz

---

> ### Comment · Reviewer_CFkx · 2023-11-22
>
> Thank you for the additional experiments. Could you please comment on why the residual loss achieved better results for the L-shaped domain? As the theory suggests, I would expect Astral to always give better energy norms.

---

> ### Author Response · Authors · 2023-11-22
>
> We can propose a few plausible explanations for that:
> 1. Astral loss is merely an upper bound on the error energy norm, not the energy norm itself. If the neural network predicts poor certificates, it can affect the quality of the solution since the upper bound will be loose.
> 2. Error norm contains derivatives, so the smoother the solution, the better. Residual loss heavily penalizes smoothness since it uses a strong form, i.e., it has a second derivative with respect to the network input. (This also means, it does not work for problems where only a weak form is available.) On the other hand, Astral loss contains only first derivatives, so the obtained solution can be less smooth.

---

### Author Response · Authors · 2023-11-22
**Summary of changes**

Following the reviewers' suggestions, we present a revised manuscript. The list of changes is as follows:

1. Section 2.3. contains one more example of Astral loss for the convection-diffusion equation.
2. In the new Section 2.4. we discuss the relation between the $L_2$ norm and the energy norm.
3. Table 1 summarizes the results of the convection-diffusion equation and the elliptic equation in the L-shaped domain (classical benchmark for methods used for a posteriori error analysis). The results include a comparison of the relative error norm and energy norm for training with residual and Astral losses, as well as the quality of the upper bound obtained with Astral loss. Details of experiments are available in two new appendices.
4. We corrected typos spotted by reviewers.
5. To make the paper more assessable, we include the discussion of a posteriori and a priori error estimates and two new appendices that contain the derivation of Astral losses for elliptic equations.
6. The reproducibility section contains a link to the anonymous version of the GitHub repository.
7. Related research is transferred to the appendix and contains one extra reference suggested in the public comment.

---

### Meta-Review · Area_Chair_Lr62 · 2023-12-16

**Metareview:**

The authors proposed a new loss function for training of neural networks that are designed to solve PDEs. This loss function is inspired by the theory of functional a posteriori error estimates. The introduction of a posteriori error estimates is interesting and can be useful for constructing new loss functions. There are indeed potentials from this work. However, despite the theory from a posteriori error estimates gives a bound for the error, which suggests the obtained solution using this loss should achieve better accuracy, the experimental results are not sufficient to demonstrate the effectiveness of this method—1) The evaluation metric often used from the literature is not computed in many experiments; 2) In some experiments, the proposed method gives a worse solution comparing to previous methods and there is no discussion on it. We thus recommend rejection. We would encourage the authors to continue their study and show more compelling empirical results. Additionally, the paper may benefits from more expository discussion about the theory of a posteriori error estimates.

**Justification For Why Not Higher Score:**

The experiments is not sufficient even after author feedback.

**Justification For Why Not Lower Score:**

N/A

---

### Decision · Program_Chairs · 2024-01-16

Reject